# Thalamocortical axons control the cytoarchitecture of neocortical layers by area-specific supply of VGF

**Haruka Sato[1]\*, Jun Hatakeyama[1], Takuji Iwasato[2], Kimi Araki[3], Nobuhiko Yamamoto[4], Kenji Shimamura[1]\***

[1]Department of Brain Morphogenesis, Institute of Molecular Embryology and Genetics, Kumamoto University, Kumamoto, Japan; [2]Laboratory of Mammalian Neural Circuits, National Institute of Genetics, Mishima, Japan; [3]Division of Developmental Genetics, Institute of Resource Development and Analysis, Kumamoto University, Kumamoto, Japan; [4]Laboratory of Cellular and Molecular Neurobiology, Graduate School of Frontier Biosciences, Osaka University, Osaka, Japan

**\*For correspondence:**
stharuka@kumamoto-u.ac.jp (HS);
simamura@kumamoto-u.ac.jp (KS)

**Competing interest:** The authors declare that no competing interests exist.

**Abstract** Neuronal abundance and thickness of each cortical layer are specific to each area, but how this fundamental feature arises during development remains poorly understood. While some of area-specific features are controlled by intrinsic cues such as morphogens and transcription factors, the exact influence and mechanisms of action by cues extrinsic to the cortex, in particular the thalamic axons, have not been fully established. Here, we identify a thalamus-derived factor, VGF, which is indispensable for thalamocortical axons to maintain the proper amount of layer 4 neurons in the mouse sensory cortices. This process is prerequisite for further maturation of the primary somatosensory area, such as barrel field formation instructed by a neuronal activity-dependent mechanism. Our results provide an actual case in which highly site-specific axon projection confers further regional complexity upon the target field through locally secreting signaling molecules from axon terminals.

## Editor's evaluation

Here Sato and colleagues in the Shimamura lab investigate the role of extrinsic factors in the development of the murine neocortex. They show that factors provided by thalamocortical afferents, namely, VGF, are instructive in the formation of layer 4 neuron populations and thus layer thickness, particularly for the primary somatosensory cortex. This study in vivo, based on VGF knockouts, expands their previous findings in vitro on this process, and adds to our understanding of how cytoarchitectonic differences across cortical areas are established.

## Introduction

Making differences within a seemingly uniform entity of cells is a fundamental process observed at various situations in development. The molecular mechanisms that underlie those processes are of great interest for not only developmental biologists but also researchers in stem cell biology and regenerative medicine. While a huge variety of systems or mechanisms have been investigated, providing the basic concepts or principles underlying this issue, there must be yet unidentified mechanisms. The adult mammalian neocortex is entirely composed of six layers of neurons, yet the laminar structure is not uniform throughout the neocortex; the thickness and cellular composition

of the layers differ among cortical areas (*Brodmann, 2006*). For instance, in the sensory cortex, layer 4, which is the main recipient layer of sensory information from the thalamus, is thick and dense, whereas it is thinner in the motor area. These features are considered to be crucial for proper functions of the cortical areas, as minute alterations in these constituents have been shown to be associated with cognitive, mental, or psychiatric disorders (*Bruining et al., 2015*; *Reavis et al., 2017*; *Selten et al., 2018*). The developmental mechanisms that regulate the formation of the regionally distinct laminar architecture therefore has long been a major issue in developmental neurobiology.

Previous studies have shown that both mechanisms intrinsic and extrinsic to the cortex play roles in the formation of cortical areas (reviewed in *Cadwell et al., 2019*). For instance the secretory factors, emanating from the signaling centers, set up the areal pattern of the neocortex, called cortical area map, through regulating the expression of transcription factors in the cortical primordium (*Armentano et al., 2007*; *Bishop et al., 2000*; *Fukuchi-Shimogori and Grove, 2001*; *O'Leary and Sahara, 2008*). While the extrinsic mechanisms are less understood, especially at the molecular level, laminar differences among cortical regions correlate well with thalamocortical axon (TCA) projection patterns: abundant TCAs project to sensory areas, which exhibit a thick layer 4, whereas only few axons project to the motor area, which has a thin layer 4. This correlation raises the possibility that TCAs may extrinsically regulate the area differences. In fact, roles of TCAs in cortical development have been investigated particularly with respect to neuronal activity-dependent mechanisms; histological and functional features called barrel in the primary somatosensory area (S1) in rodents and ocular dominance column in the primary visual area (V1) of cats and monkeys are instructed by the activity-dependent mechanisms (*Penn and Shatz, 1999*; *Katz and Crowley, 2002*; *Gaspar and Renier, 2018*; *Martini et al., 2018*). Recent studies demonstrated that TCAs also play instructive roles in the specification of area properties of somatosensory and visual cortices represented by expression of area markers (*Chou et al., 2013*; *Pouchelon et al., 2014*; *Vue et al., 2013*) as well as layer markers, and morphogenesis of cortical neurons (*Li et al., 2013*; *Zhou et al., 2010*). Although molecular mechanisms underlying those processes remain uncovered, it was reported that prenatal thalamic neuronal activities and propagation of calcium waves regulate cortical maps prior to sensory processing (*Antón-Bolaños et al., 2019*; *Moreno-Juan et al., 2017*). Several studies have also shown that TCA innervation influences neurogenesis in the embryonic cortex. For example, ephrin A5 expressed in TCAs regulates the generation of proper types of cortical progenitor cells and thus neuronal output for cortical layers (*Gerstmann et al., 2015*). Wnt3 secreted by TCAs controls neuronal differentiation in the cortex at the translational level (*Kraushar et al., 2015*). However, whether TCAs regulate cytoarchitectural aspects of the cortical layers such as the number of cortical neurons and layer thickness is not clear. Only one study suggested the involvement of TCAs in cortical laminar formation by showing that thalamic ablation by electrolytic lesion led to alterations in the cortical laminar configuration (*Windrem and Finlay, 1991*). Yet, it is technically difficult to target thalamic nuclei specifically and reproducibly by surgical manipulation, and the molecular mechanisms by which TCAs control cortical laminar organization have not been uncovered.

To identify TCA-derived extrinsic factors, we previously conducted a screening for thalamus-specific genes by comparing expression profiles of the thalamus and the cortex (*Sato et al., 2012*). As a result, two genes encoding neuritin 1 (NRN1) and VGF nerve growth factor inducible (VGF) were found to be expressed specifically in sensory thalamic nuclei including the ventrobasal nucleus (VB). While their mRNAs are not expressed in the cortex, their proteins are detected in cortical layer 4, suggesting that they are transported to the cortex through TCAs. We further found that NRN1 and VGF promoted survival and dendrite growth of cortical neurons in vitro (*Sato et al., 2012*). Although these extrinsic factors are likely to contribute to cortical development in vivo, this remains to be validated.

In this study, we investigated the effect of loss of TCAs on neocortical development using transgenic mice, in which thalamic neurons were eliminated postnatally. We employed a diphtheria toxin (DT) receptor (DTR)-mediated cell ablation system combined with a thalamus-specific Cre transgenic mouse line (*Arakawa et al., 2014*; *Buch et al., 2005*). As a result, the laminar structure was altered specifically in layer 4, which exhibited marked reduction in the number of neurons in the primary somatosensory cortex. Moreover, TCA-derived factor VGF is necessary for the proper amount of layer 4 neurons in S1 and V1. Interestingly, the barrel organization was impaired in *Vgf*-knockout mice, despite the presence of TCAs and their activities, suggesting that the VGF-dependent quantity

control is crucial for the proper barrel field development in cooperation with the activity-dependent process.

## Results

### Distinctions in laminar configuration develop postnatally

We first determined when the laminar differences among areas emerge during mouse cortical development by analyzing the expression of RAR-related orphan receptor beta (RORβ), a layer 4 marker, across the cortical areas in a sagittal plane from embryonic to postnatal stages using an anti-RORC antibody (see Materials and methods). At E16 when the production of layer 4 neurons is completed, RORβ expression was high in the anterior and low in the posterior regions (*Figure 1—figure supplement 1A*, in three mice). RORβ expression then became relatively uniform throughout the cortex, with slight fluctuation in intensity at birth (*Figure 1—figure supplement 1A*, in three mice). By P7, the staining intensity and thickness of the RORβ-expressing cell layer became greater in the primary somatosensory cortex (S1; *Figure 1—figure supplement 1A*, in five mice). In a coronal plane, RORβ was expressed in a dorsal low-ventral high gradient from E16 to P0 (*Figure 1—figure supplement 1B*, in five mice for both stages). By P7, a thick and densely stained domain became evident, which is recognized as posteromedial barrel subfield in S1 (*Figure 1—figure supplement 1B*, in five mice). This observation was also confirmed quantitatively (*Figure 1—figure supplement 1C*). These data indicated that area-specific laminar characteristics are formed postnatally.

Next, we asked what mechanisms regulate this transition. As neurogenesis in the cortex is nearly completed at the time of birth (*Kwan et al., 2012*), it is unlikely that layer 4 neurons are additively generated in S1. Indeed, when EdU was administered every day from P0 to P4, to label newly generated cells during this period, EdU-labeled cells in layer 4 were all negative for NeuN at P6 (*Figure 1—figure supplement 2*, in 16 sections from 4 mice), indicating that layer 4 neurons were not newly generated during this period. Thus, we concluded that neurogenesis does not take place to form cortical layers in postnatal stages. On the other hand, after arriving at the cortex during the embryonic stages, TCAs start innervating cortical layer 4 at the postnatal stages, temporally correlating with the development of the laminar configuration (*López-Bendito and Molnár, 2003*).

### Neonatal ablation of TCAs projecting to S1

To explore the role of TCA projection in layer formation, sensory thalamic neurons were ablated using the Cre-inducible DTR mouse system (*Buch et al., 2005*). A 5HTT-Cre transgenic mouse, in which Cre recombinase activity was detected in sensory thalamic neurons by crossing with an R26-EYFP reporter mouse (*Srinivas et al., 2001*; *Figure 1A*, in five mice), was crossed with an R26-DTR mouse (*Figure 1B*). DT was then administrated at P0, and the brains were analyzed at P5–7. At P5, we observed numerous dying cells, detected by ssDNA staining (data not shown) and accumulation of Iba1-immunoreactive microglial cells, which scavenge dead cells, in the thalamus (*Figure 1C*, in six sections from six mice), consistent with the previous study (*Arakawa et al., 2014*). Thalamic cell ablation was further evaluated by immunohistochemistry with an anti-RORC antibody. The VB was dramatically degenerated at P6 (size reduced by 67.2% compared with control), and RORα+β-expressing thalamic neurons were considerably decreased in the VB where Cre expression was the highest among the thalamic nuclei in 5HTT-Cre mice (*Figure 1D, E*, in seven mice). On the other hand, the size of the dorsal lateral geniculate nucleus (dLGN) was not obviously reduced, and RORα+β-expressing cells were detected (*Figure 1D, E*, in seven mice). Consequently, the terminals of TCAs were severely diminished in S1 as revealed by 5HTT immunostaining (*Figure 1F*). Hereafter, we refer to these animals as TCA-ablated mice and animals with DT administration, but without 5HTT-Cre allele, are referred to as control mice.

To examine further whether TCAs are eliminated or neural connections are altered in TCA-ablated mice, the lipophilic dyes DiA and DiI were injected into two different TCA target areas, S1 and the primary visual cortex (V1), respectively, to label TCAs and thalamic neurons retrogradely (*Figure 1—figure supplement 3A*, in 12 sections from 6 mice); S1 and V1 are the major target of VB and dLGN, respectively, and the retrogradely labeled neuronal cell bodies could be distinguished easily from diffusely labeled cortical afferents in the thalamus by the fluorescent intensity. In control animals, a large number of DiA-labeled thalamic neurons were detected in the VB and the posterior nucleus (PO) (*Figure 1—figure supplement 3B*, 12 sections from 6 mice). In contrast, in TCA-ablated mice,

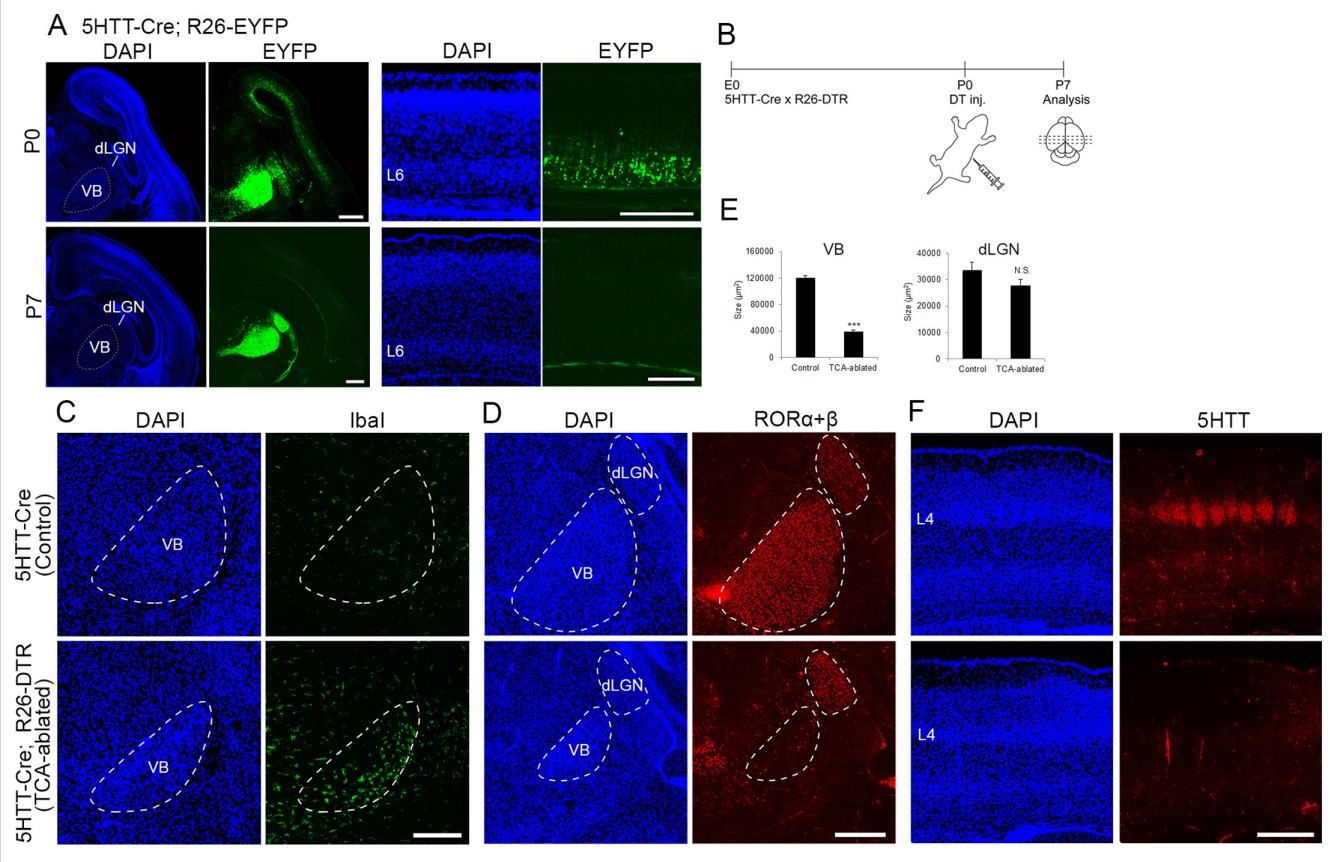

**Figure 1.** Elimination of thalamocortical axons (TCAs) by toxin-mediated thalamic cell ablation in vivo. (**A**) Cre recombinase activities in cross-sections of forebrains of P0 (upper panels) and P7 (lower panels) 5HTT-Cre mice. The 5HTT-Cre line was crossed with the R26-EYFP reporter line to allow detection of Cre activity by EYFP fluorescence. EYFP was the most strongly expressed in the VB nucleus and to a lesser extent in the dLGN in the thalamus at P0 (upper left panels), and also weakly in layer 6 (L6) in the cortex (upper right panels), which was lost by P7 (lower right panels). (**B**) Experimental scheme for TCA ablation. Neonatal pups from a cross of a 5HTT-Cre mouse with a R26-Diphtheria toxin receptor (DTR) mouse were administrated DT intraperitoneally. Cross-sections of P5 (**C**) and P6 (**D**) thalamus stained for Iba1 (**C**) and RORα+β (**D**) reveal microglial cells and thalamic nuclei, respectively. Ablation of thalamic neurons was confirmed by loss of RORα+β expression, especially in the VB. DAPI (4',6-diamidino-2-phenylindole)-stained cell nuclei are packed more densely in the smaller VB than in the control. (**E**) Quantification of the size of VB and dLGN. The area of these nuclei revealed by RORC staining on seven sections from seven mice was measured. The size of VB, but not dLGN was significantly reduced in TCA-ablated mice compared with control mice. Data are presented as μm$^2$ (mean ± standard error of the mean [SEM]): VB, 120,882.2 ± 5582.1 (control), 39,658.6 ± 5672.2 (TCA-ablated), p = 0.0006; dLGN, 33,514.1 ± 3081.1 (control), 27,699.9 ± 2308.1 (TCA-ablated), p = 0.128; Mann–Whitney $U$ test, ***p < 0.001. Seven mice for both control and experiment were used. (**F**) 5HTT immunohistochemistry of P7 cortices. The amount of TCAs stained for 5HTT was greatly reduced in S1, the target of VB neurons, in the TCA-ablated cortex. Note that the 5HTT-positive barrel structure is completely absent in TCA-ablated mice. dLGN, dorsal lateral geniculate nucleus; L4, layer 4; L6, layer 6; VB, ventrobasal nucleus. Scale bars, 500 μm (**A, F**), 200 μm (**C, D**).

The online version of this article includes the following source data and figure supplement(s) for figure 1:

**Source data 1.** Raw data of E.

**Figure supplement 1.** Postnatal emergence of laminar configuration in the cortical area.

**Figure supplement 1—source data 1.** Raw data of C.

**Figure supplement 2.** Layer 4 neurons are not additively generated postnatally.

**Figure supplement 3.** Thalamocortical axon (TCA) connections in S1 of TCA-ablated mouse.

the retrogradely labeled VB was markedly reduced in size, but the PO was broadly and intensely labeled (*Figure 1—figure supplement 3B*, 12 sections from 6 mice). On the other hand, DiI-labeled neurons that project to V1 were found in the dLGN and lateral posterior nucleus (LP) in both control and TCA-ablated mice (*Figure 1—figure supplement 3A, B*, 12 sections from 6 mice for each). Moreover, the fact that DiA-labeled S1-projecting cells were not detected in the dLGN (*Figure 1—figure supplement 3B*, 12 sections from 6 mice) and the medial geniculate nucleus (MG) (*Figure 1—figure*

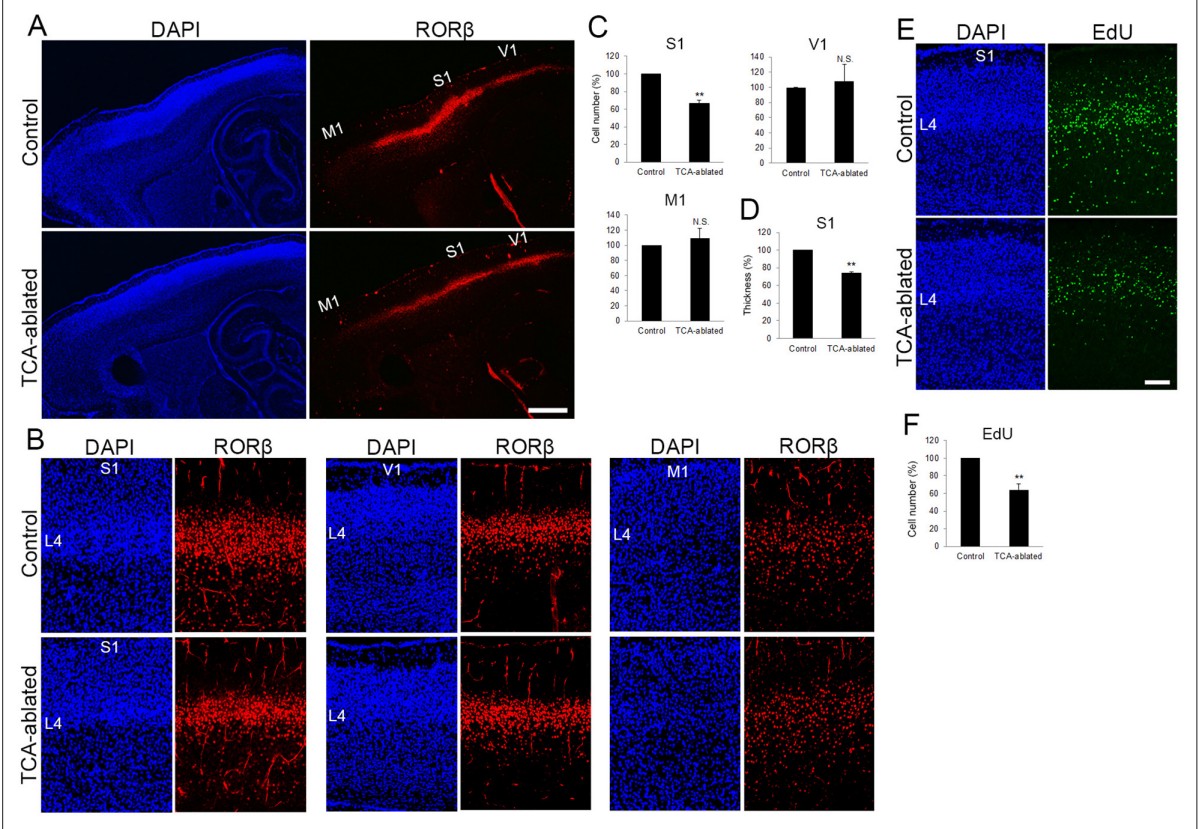

**Figure 2.** The number of layer 4 neurons is reduced in S1. (**A**) Sagittal sections of P7 cortices derived from thalamocortical axon (TCA)-ablated mice stained with RORC antibody. Expression of the layer 4 marker RORβ was declined, specifically in S1, resulting in poorly demarcated borders between adjacent areas. (**B**) Coronal section of the cortex of a TCA-ablated mouse at P7, showing RORβ-expressing layer 4 (L4) neurons in S1. (**C**) Quantification of RORβ expression. RORβ-expressing cells within an 850-μm-wide strip of the cortical wall were counted. Note that the number of RORβ-positive cells was less in S1, but was not changed in V1 and M1, in TCA-ablated mice. Data are presented as a percentage of control mice (mean ± standard error of the mean [SEM]): S1, 66.75 ± 3.30%, $N$ = 7 mice, p = 0.0011; V1, 108.28 ± 22.00%, $N$ = 5 mice, p = 0.656; M1, 109.47 ± 12.97%, $N$ = 4 mice, p = 0.878; Mann–Whitney $U$ test, **p < 0.01. The same numbers of control and experimental animals were used. (**D**) Thickness of RORβ-expressing layer was also reduced in TCA-ablated mice. Data are presented as a percentage of control mice (mean ± SEM): 73.69 ± 1.83%, $N$ = 7 mice, p = 0.0011; Mann–Whitney $U$ test, **p < 0.01. (**E**) Cross-sections of S1 cortices of control and TCA-ablated mice at P7, showing distribution of EdU-positive cells. EdU was injected at E14.7. (**F**) Quantification of the results. The number of EdU-positive cells within an 850-μm-wide strip of the cortical wall relative to control is shown as the mean ± SEM: 64.37 ± 6.77%, p = 0.0075; Mann–Whitney $U$ test, **p < 0.01, $N$ = 5 animals for both control and TCA-ablated mice. L4, layer 4; M1, motor area; S1, primary somatosensory area; V1, primary visual area. Scale bars, 500 μm (**A**), 100 μm (**B, E**).

The online version of this article includes the following source data and figure supplement(s) for figure 2:

**Source data 1.** Raw data of C, D, F.

**Figure supplement 1.** Effects of thalamocortical axon (TCA) ablation on the number of neurons in L2–5 and Cux1-positive upper layer neurons.

**Figure supplement 1—source data 1.** Raw data of C.

**Figure supplement 2.** Effects of thalamocortical axon (TCA) ablation on cell fate and cell death of layer 4 neurons in S1.

**Figure supplement 2—source data 1.** Raw data of B, D.

**Figure supplement 3.** Electroporation-based thalamocortical axon (TCA) ablation causes reduction in the number of layer 4 neurons.

*supplement 3C*, 12 sections from 6 mice) suggests that TCA rewiring from the dLGN or MG to S1 did not occur in TCA-ablated mice, unlike previously reported findings (*Mezzera and López-Bendito, 2016*). Thus, S1 in TCA-ablated mice receives very few axonal inputs from the sensory thalamic nuclei including VB, dLGN, and MG.

## The number of layer 4 cells in S1 is reduced in TCA-ablated mice

To examine impacts of TCA ablation on the cortical laminar structure, the number of cells in each layer was analyzed in control and TCA-ablated cortex using layer makers. RORC immunostaining revealed

that layer 4 of S1 was thinner in TCA-ablated than in control mice, leading to poor demarcation of S1 (*Figure 2A, B*, in five sections from three mice for each). Quantitative analysis revealed that the number of RORβ-expressing cells was decreased by 33% as compared with control cortex (*Figure 2C*, 15 sections from 7 mice). However, such reduction in layer 4 cells was not observed in nontarget areas of VB axons, M1 and V1. V1 still received TCA projection from the remaining dLGN neurons in the experimental situation as described above (*Figure 2C*, V1, eight sections from five mice; M1, five sections from four mice; see *Figure 1D, E*). Concomitantly, the thickness of the layer abundant in RORβ-expressing cells was significantly reduced in TCA-ablated mice (by 26% compared with control; *Figure 2D*, 16 sections from 7 mice).

To examine whether the reduction of RORβ-expressing cells in TCA-ablated mice is due to a reduction of cells that had been destined for layer 4, we labeled postmitotic neurons by injecting EdU into the mother at E14.3 or E14.7 (see Materials and methods), when most layer 4 neuron progenitors are in S-phase before terminal mitosis. As expected, EdU-positive cells were preferentially distributed in layer 4 at P7 (*Figure 2E*, in 12 sections from 5 mice). The total number of EdU-positive cells was significantly decreased in TCA-ablated S1 (64% of that in the control; *Figure 2E, F*, 12 sections from 5 mice for each). The extent of reduction was comparable to that of RORβ-expressing cells (*Figure 2C*). Although the total number of NeuN-positive neurons in layers 2–5 was not substantially different (*Figure 2—figure supplement 1A, C*), the number of Cux1-positive upper layer neurons including layer 4 turned out to be slightly decreased in TCA-ablated animals (*Figure 2—figure supplement 1B, C*). To examine further the cell fate of layer 4-destined cells in TCA-ablated mice, the proportion of EdU-positive cells that express each layer-specific marker was analyzed. The percentage of RORβ-expressing cells was not markedly different between control and TCA-ablated mice (*Figure 2—figure supplement 2A, B*, 26 sections from 7 mice for each). The proportion of other layer marker-expressing cells (Brn2 for layer 2/3 and Ctip2 for layer 5) to EdU-positive cells were not substantially increased in TCA-ablated mice compared to control mice (*Figure 2—figure supplement 2A, B*, 15 sections from 5 mice for each condition of Brn2; 14 sections from 4 mice for each condition of Ctip2). In fact, there was no increase in EdU-positive cells in other layers (*Figure 2E*, *Figure 2—figure supplement 2A*, 12 sections from 5 mice). Taken together, these results suggest that the absolute number of layer 4 neurons decreased in the TCA-ablated S1, and argue against the possibilities that altered RORβ expression or fate change of layer 4-destined cells to those of other layers is the major cause of the reduction of RORβ-positive cells.

Next, we examined whether cell death is involved in the reduction in layer 4 cells in S1. As 5HTT-Cre is expressed in VB and cortical layer 6 (see *Figure 1A*), cell death was induced in both regions within several days after DT administration at P0 (*Figure 2—figure supplement 2C, D*). However, although we used several cell death detection methods (i.e., ssDNA, cleaved caspase 3, Iba1, mRNA of *Bax*, *Bad*, and *Bak*, and DAPI), we could not obtain convincing evidence for significant cell death induction in layer 4 upon TCA ablation (*Figure 2—figure supplement 2C, D*, 12 sections from 3 mice for P1 control; 8 sections from 2 mice for P1 TCA-ablated; 12 sections from 3 mice for P2; 11 sections from 3 mice for P3; 9 sections from 3 mice for P4 and P5; 13 sections from 4 mice for P6; 16 sections from 4 mice for P7). Given the technical difficulties in detecting dead or dying cells in postnatal brain due to the rapid clearance of dead cells (*Gohlke et al., 2004*; *Wong et al., 2018*), it is still possible that cell death is involved in the reduction of layer 4 in TCA-ablated mice.

As Cre recombinase is active not only in the thalamus, but also in cortical layer 6 at P0 (*Figure 1A*, 10 sections from 5 mice) and the raphe nucleus (data not shown; *Arakawa et al., 2014*) in 5HTT-Cre mice, we cannot exclude that the layer 4 reduction in 5HTT-Cre; R26-DTR mice is due to ablation of these brain parts rather than the VB in the thalamus. To verify that TCA ablation was indeed responsible for the laminar phenotype of 5HTT-Cre; R26-DTR mice, we ablated VB neurons using a different approach in which a DTR expression plasmid was electroporated into the embryonic dorsal thalamus in utero at E11.5, when VB neurons are generated (*Figure 2—figure supplement 3A, B*, five sections from five mice). Despite variable transfection efficiency in the thalamus, DT administration at P0 induced massive cell death in the VB (*Figure 2—figure supplement 3C*, eight sections from four mice) and led to a reduction in cell density in the VB (*Figure 2—figure supplement 3D*, seven sections from five mice). Moreover, 5HTT-positive axon terminals in the cortex were severely reduced (*Figure 2—figure supplement 3E*, five sections from three mice), and the number of layer 4 neurons in S1 was decreased (*Figure 2—figure supplement 3F*, eight sections from five mice for

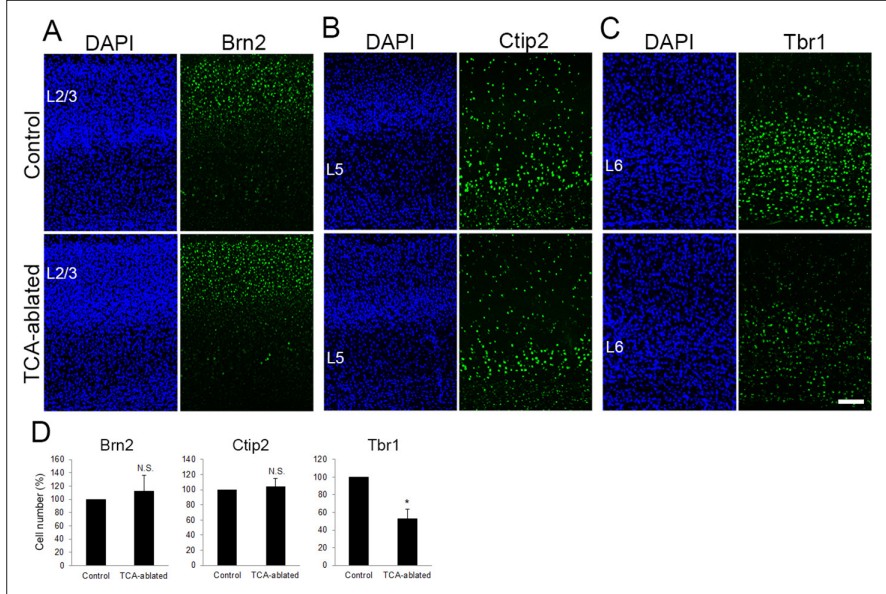

**Figure 3.** Layers 2/3 and 5 appear intact upon thalamocortical axon (TCA) elimination. Cross-sections of S1 cortices of TCA-ablated mice at P7 stained for Brn2 (layer 2/3) (**A**), Ctip2 (layer 5) (**B**), and Tbr1 (layer 6) (**C**). Expression of Brn2 and Ctip2 were not markedly changed, whereas Tbr1-positive cells were decreased in TCA-ablated mice. (**D**) Quantitative analyses of the number of cells of TCA-ablated S1 relative to the control. Cells positive for each marker within an 850-μm-wide strip of the cortical wall were counted. Data are presented as a percentage of control (mean ± standard error of the mean [SEM]): Brn2, 112.70 ± 23.79%, $N = 6$ animals, $p = 0.347$; Ctip2, 104.09 ± 11.23%, $N = 4$ animals, $p = 0.878$; Tbr1, 52.97 ± 10.47%, $N = 4$ animals, $p = 0.0211$; Mann–Whitney $U$ test, *$p < 0.05$. The same numbers of control and experimental animals were used. L2/3, layers 2 and 3; L5, layer 5; L6, layer 6. Scale bar, 100 μm.

The online version of this article includes the following source data for figure 3:

**Source data 1.** Raw data of D.

each), reminiscent of the 5HTT-Cre; R26-DTR phenotype (**Figure 2B**). We also examined whether cell death in layer 6 which occurs in 5HTT-Cre; R26-DTR mice affects layer 4 neurons. The DTR expression plasmid was electroporated to E11.5 cortices when layer 6 neurons are born and DT was administrated at P0. We found reduction of cell number in layer 6, but not in layer 4 (data not shown). Taken together, these results strongly supported the notion that the reduction in layer 4 cells was caused by the loss of TCAs.

## Layers 2/3 and 5 are intact in TCA-ablated mice

To examine whether other layers were also affected by TCA ablation, expression of Brn2 (layers 2/3), Ctip2 (layer 5), and Tbr1 (layer 6) was analyzed. The number of Brn2-positive cells was not markedly changed in TCA-ablated cortex (**Figure 3A, D**, 17 sections from 6 mice for each). Similarly, the number of Ctip2-expressing cells was not greatly affected (**Figure 3B, D**, 17 sections from 4 mice for each). In contrast, Tbr1-expressing cells were partially reduced in TCA-ablated mice (**Figure 3C, D**, 10 sections from 4 mice for each), likely due to DT-induced cell death of the layer 6 neurons that express Cre recombinase at P0 as mentioned above (**Figure 1A**). Indeed, we observed signs of cell death in layer 6 (i.e., ssDNA-positive, accumulation of microglia; **Figure 2—figure supplement 2D**, data not shown) in the TCA-ablated S1. Collectively, these results supported the specificity of the effects of TCA ablation on cortical layer formation in that the effect was restricted to the target layer of TCA projection.

## The number of layer 4 neurons is restored by forced expression of VGF in the cortex of TCA-ablated mice

Regarding the molecular basis of TCA-dependent layer 4 formation, we hypothesized that extracellular molecules emanating from TCA terminals are involved. As our previous study showed that NRN1 and VGF which promote cortical cell survival and dendritic growth are localized in TCA terminals

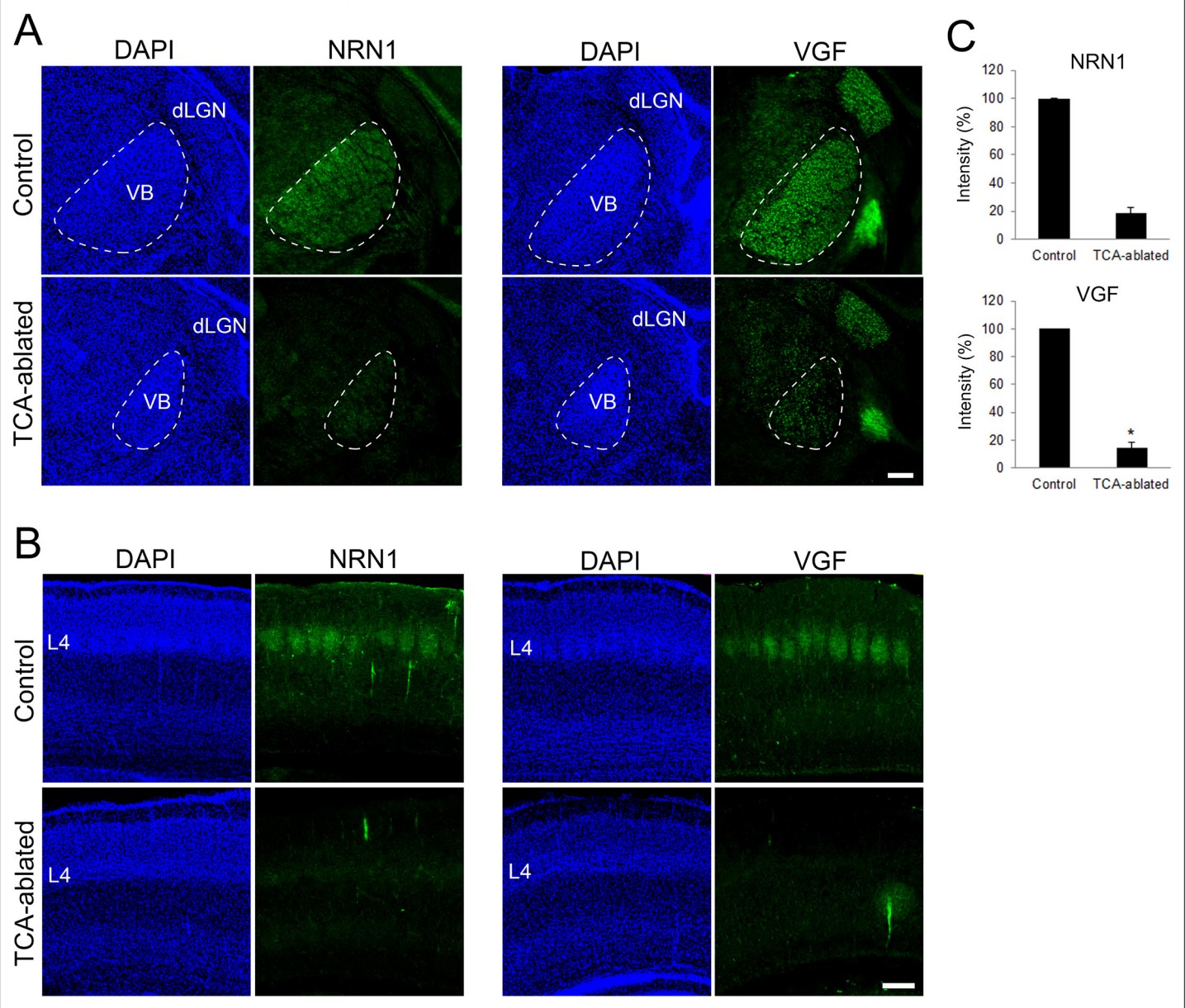

**Figure 4.** Expression of NRN1 and VGF are lost in the cortex of thalamocortical axon (TCA)-ablated mice. Coronal sections of thalamus (**A**) and S1 cortex (**B**) of control and TCA-ablated mice at P7 immunostained for NRN1 and VGF. Note that NRN1 and VGF are expressed in the VB of control, but not TCA-ablated mice, and that their signals are absent in layer 4 of S1 cortex in TCA-ablated mice. (**C**) Quantification of the intensity of NRN1- and VGF-immunoreactive signal in the VB. Data are presented as a percentage of control (mean ± standard error of the mean [SEM]): NRN1, 18.35 ± 4.03 %, *N* = 2 for each; VGF, 13.43 ± 2.85%, *N* = 4 for control, *N* = 5 for TCA-ablated mice, p = 0.0151; Mann–Whitney *U* test, *p < 0.05. dLGN, dorsal lateral geniculate nucleus; L4, layer 4; VB, ventrobasal nucleus. Scale bar, 200 μm.

The online version of this article includes the following source data for figure 4:

**Source data 1.** Raw data of C.

(*Sato et al., 2012*), the expression and action of these molecules were investigated. As expected, the expression of both NRN1 and VGF was lost in the thalamic nuclei and their axon terminals in layer 4 of S1 in TCA-ablated mice (*Figure 4A–C*; NRN1, two sections from two mice for each; VGF, four sections from four mice for control, five sections from five mice for TCA-ablated). To examine whether these proteins play any role for the number of layer 4 neurons in S1, we overexpressed these factors in layer 4 cells by in utero electroporation prior to TCA ablation (*Figure 5A, B*, six mice; *Figure 5—figure supplement 1A, B*, six sections from three mice). As a result, the number of layer 4 cells defined by

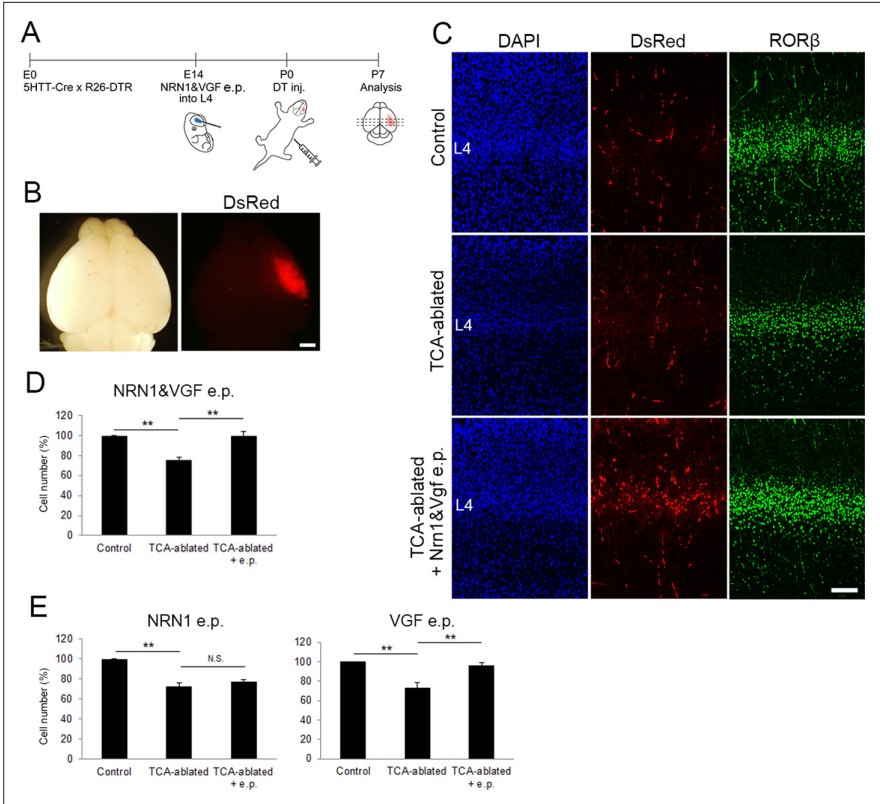

**Figure 5.** Overexpression of VGF rescues layer 4 formation in thalamocortical axon (TCA)-ablated mice. (**A**) Schematic representation of the experimental procedure. (**B**) P7 electroporated (e.p.) brain showing the DsRed-positive e.p. region in the right hemisphere. (**C**) Cross-sections of S1 cortices of control, TCA-ablated, and TCA-ablated + e.p. mice stained for DsRed and RORβ. (**D**) Quantitative analysis of RORβ-expressing cells. Data are presented as a percentage of control (mean ± standard error of the mean [SEM]): TCA-ablated, 75.61 ± 2.87%, p = 0.0028; TCA-ablated + e.p., 99.52 ± 4.54%, p = 0.0022, N = 6 animals for each; Mann–Whitney U test, **p < 0.01. (**E**) Results of single electroporation of NRN1 or VGF into layer 4 of TCA-ablated mice. The number of RORβ-expressing cells was counted and presented as a percentage of control (mean ± SEM): NRN1 e.p., 72.27 ± 3.68% (TCA-ablated), p = 0.0075, 76.87 ± 2.18% (TCA-ablated +e.p.), p = 0.5476, N = 5 animals for each; VGF e.p., 73.43 ± 4.91% (TCA-ablated, N = 5 animals for control, 7 animals for TCA-ablated), p = 0.0042, 96.2 ± 3.10% (TCA-ablated + e.p., N = 5 animals for control, 7 animals for TCA-ablated), p = 0.0023; Mann–Whitney U test, **p < 0.01. L4, layer 4. Scale bars, 1 mm (**B**), 100 μm (**C**).

The online version of this article includes the following source data and figure supplement(s) for figure 5:

**Source data 1.** Raw data of D.

**Figure supplement 1.** Expression of exogenous NRN1 and VGF in the electroporated cortices.

**Figure supplement 2.** Overexpression of NRN1 and VGF did not affect layer 4 formation in control mice.

**Figure supplement 2—source data 1.** Raw data of C.

RORC immunoreactivity was completely restored to the control level in TCA-ablated mice (*Figure 5C, D*, 11 sections from 6 animals for each). Curiously however, we did not observe an additive increase in layer 4 neurons by overexpression of these factors in the presence of TCA (*Figure 5—figure supplement 2B*, 11 sections from 6 mice for each). To determine which factor is responsible for the restoration of layer 4 neurons, we further performed in utero electroporation of either NRN1 or VGF into layer 4. Whereas NRN1 overexpression did not restore the reduction of RORC-immunoreactive layer 4 cells (*Figure 5E*, 18 sections from 5 mice for each), VGF overexpression was capable of rescuing the layer 4 phenotype in TCA-ablated mice (*Figure 5E*, 18 sections from 5 mice for control, 29 sections from 7 mice for TCA-ablated). Taken together, these results suggested VGF as a TCA-derived factor to maintain the layer 4 neuronal number during postnatal neocortical development.

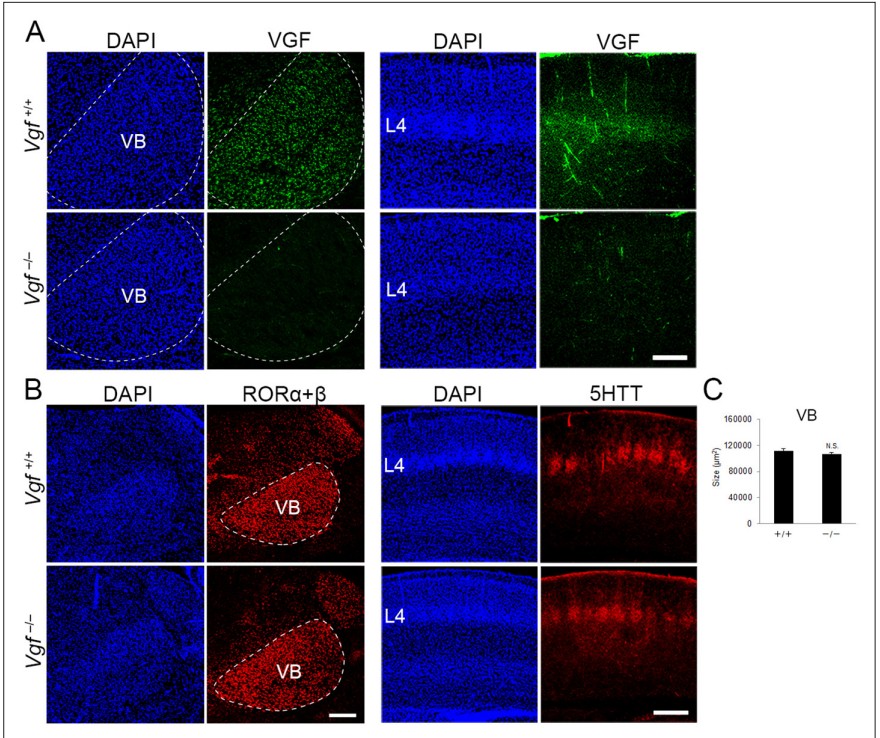

**Figure 6.** Generation of *Vgf*-KO mice using the CRISPR/Cas9 system. Coronal sections of thalamus (left groups) and S1 cortex (right groups) of wild-type and *Vgf*-deficient mice at P8 stained with anti-VGF (**A**), -RORC (**B**), and -5HTT (**B**) antibodies. (**C**) Quantification of the size of the VB nucleus: 111,800.71 ± 3366.63 μm² (wild-type), 106,920 ± 2667.67 μm² (*Vgf*-KO), *N* = 8 mice for both, p = 0.235, Mann–Whitney *U* test. Note that VGF protein is lost in the VB and cortical layer 4 of *Vgf*⁻/⁻ mice, whereas expression of RORα+β in the VB and the presence of 5HTT-positive thalamocortical axon (TCA) terminals in cortical layer 4 were not affected. Scale bars, 200 μm (**A** and **B**, left panel), 500 μm (**B**, right panel).

The online version of this article includes the following source data for figure 6:

**Source data 1.** Raw data of C.

## Genetic inactivation of *Vgf* results in a reduction in layer 4 neurons in S1

We further addressed whether NRN1 and VGF are in fact necessary for the regulation of layer 4 development in S1 by using CRISPR/Cas9-mediated gene editing (*Harms et al., 2014*). We designed three single-guide RNAs (sgRNAs) cutting exons of *Nrn1* and *Vgf* to induce frame shifts resulting in failure of protein translation of both NRN1 and VGF. By electroporating the sgRNAs and Cas9 protein into fertilized eggs, mutations were induced in the genomic sequences of *Nrn1* and *Vgf* allele near designed sgRNAs. While we could not obtain double-null mice, probably due to lethality during embryonic or early postnatal stages, single-mutant mice for *Nrn1* or *Vgf*, both of which harbored null mutations, were collected at P8. We confirmed the loss of NRN1 and VGF protein by western blotting (data not shown) and immunohistochemistry (*Figure 6A*, five sections from five mice; data not shown for NRN1). Immunoreactivity of VGF was completely abolished in the VB in the thalamus, where it is endogenously expressed as described previously (*Sato et al., 2012*), as well as in layer 4 of S1, where TCAs from the VB terminate. As reported previously (*Hahm et al., 1999*), the body weight of *Vgf*⁻/⁻ mice was significantly lower than that of *Vgf*⁺/⁺ control mice (*Figure 7—figure supplement 1B*, wild-type, four mice; *Vgf*⁻/⁻, five mice). Although the size of the cortex was slightly larger, gross brain anatomy was not markedly different (*Figure 7—figure supplement 1A, B*, wild-type, four mice; *Vgf*⁻/⁻, five mice). Importantly, the thalamic structure was not affected: the VB appeared intact in terms of size and cell density (*Figure 6B, C*, eight sections from eight mice). In addition, TCAs revealed by 5HTT immunohistochemistry were present in layer 4 of S1, similar to the wild-type (*Figure 6B*, six sections from three mice), indicating that thalamocortical projection was formed properly. The effect of loss of

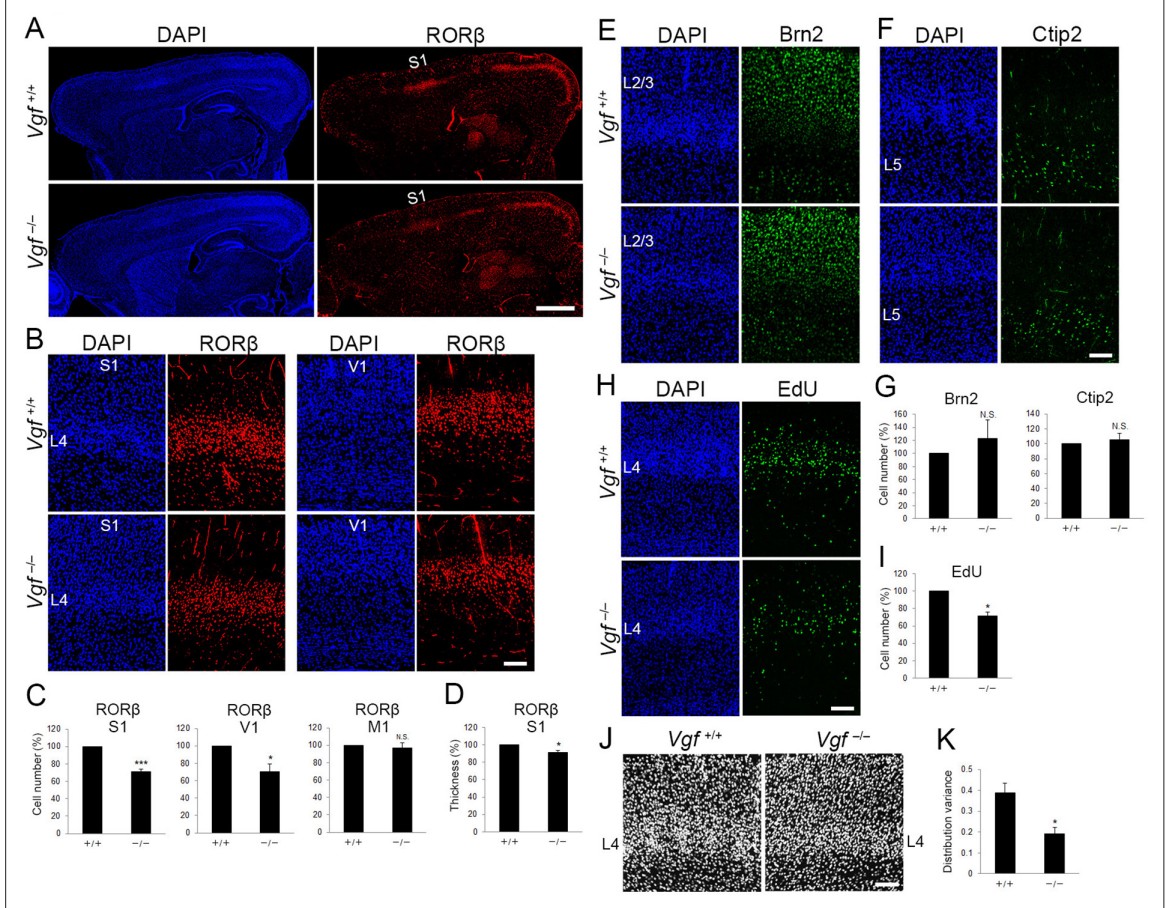

**Figure 7.** *Vgf*-deficiency causes a reduction of layer 4 neurons in S1. (**A**) RORC immunostaining of sagittal sections of P8 cortices of wild-type and *Vgf*⁻/⁻ mice revealed the reduction in layer 4 in the S1 area in the mutant. Coronal sections of P8 cortices of wild-type and *Vgf*⁻/⁻ mice stained with anti-RORC (**B**), anti-Brn2 (**E**), and anti-Ctip2 (**F**) antibodies. (**C, G**) Quantification of the results. Data are presented as a percentage of wild-type control (mean ± standard error of the mean [SEM]): RORβ S1, 70.87 ± 3.18%, $N$ = 7 wild-type and 9 *Vgf*⁻/⁻ mice, p = 0.0006; RORβ V1, 70.02 ± 9.36%, $N$ = 3 wild-type and 5 *Vgf*⁻/⁻ mice, p = 0.0314; Brn2, 123.18 ± 28.63%, $N$ = 7 wild-type and 9 *Vgf*⁻/⁻ mice, p = 0.301; Ctip2, 105.02 ± 9.37%, $N$ = 7 wild-type and 9 *Vgf*⁻/⁻ mice, p = 0.740; Mann–Whitney $U$ test, *p < 0.05, ***p < 0.001. (**D**) Relative thickness of RORβ-expressing layer to the wild-type: 91.34 ± 2.24%, $N$ = 5 *Vgf*⁻/⁻ mice, p = 0.0325; Mann–Whitney $U$ test, *p < 0.05. Three wild-type mice were used as the reference. (**H**) Cross-sections of S1 cortices of wild-type and *Vgf*⁻/⁻ mice at P7 stained for EdU. EdU was injected at E14.3. (**I**) Quantification of the results. The number of EdU-positive cells within an 850-µm-wide strip of the cortical wall relative to control is shown as the mean ± SEM: 71.22 ± 5.06%, $N$ = 4 animals for each, p = 0.0211; Mann–Whitney $U$ test, *p < 0.05. (**J**) DAPI staining of coronal sections showed barrel structure as uneven and repetitive distribution of cell nuclei in layer 4 of wild-type, but not of *Vgf*⁻/⁻ mice. (**K**) Distribution variance of layer 4 cells as the mean ± SEM: wild-type, 0.390 ± 0.045, $N$ = 4 mice; *Vgf*⁻/⁻, 0.192 ± 0.032, $N$ = 5 mice, p = 0.0159; Mann–Whitney $U$ test, *p < 0.05. L2/3, layers 2 and 3; L4, layer 4; L5, layer 5; S1, primary somatosensory area; V1, primary visual area. Scale bars, 1 mm (**A**), 100 µm (**B, E, F, H, J**).

The online version of this article includes the following source data and figure supplement(s) for figure 7:

**Uncited Figure 7—source data 1.** Raw data of C, D, G, I, K.

**Figure supplement 1.** Neonatal cell number and postnatal cell fate of layer 4 neurons in *Vgf*-KO mice.

**Figure supplement 1—source data 1.** Raw data of B, D, F, H.

**Figure supplement 2.** *Nrn1*-KO mice show normal layer 4.

**Uncited Figure 7—figure supplement 2—source data 1.** Raw data of B.

TCA-derived VGF from the cortex was evaluated by RORC immunohistochemistry. The number of the positive cells in layer 4 was markedly reduced in S1 and in V1, but not in M1 (***Figure 7A–C***, 19 sections from seven wild-type and 23 sections from 9 *Vgf*⁻/⁻ mice for S1, 4 sections from three wild-type and 8 sections from 5 *Vgf*⁻/⁻ mice for V1, 4 sections from 4 mice for both for M1). Neither layer 2/3 nor layer 5 was affected (***Figure 7E–G***, 28 sections from seven wild-type and 36 sections from 9 *Vgf*⁻/⁻ mice for Brn2, 22 sections from seven wild-type and 31 sections from 9 *Vgf*⁻/⁻ mice for Ctip2). In spite of the

significant decrease in layer 4 cells, layer thickness was less affected than the case of TCA ablation (91.3%, *Figure 7D*, four sections from three wild-type and eight sections from five *Vgf*⁻/⁻ mice; see *Figure 2D*). These findings suggested that TCAs regulate two aspects of area-specific layer 4 formation separately, such that layer thickness is controlled by the presence of TCAs, and the number of layer 4 neurons is regulated by TCA-derived VGF.

To examine whether the reduction of RORC-immunoreactive layer 4 cells in *Vgf*⁻/⁻ mice is caused by reduction of or fate change of layer 4-destined cells, this cohort was labeled by EdU at E14.3 as in the case of TCA-ablated mice (*Figure 7H*, 16 sections from 4 mice for each). First, the number of EdU-positive cells distributed preferentially in layer 4 was reduced in *Vgf*⁻/⁻ mice (*Figure 7H, I*, 16 sections from 4 mice for each), indicating that layer 4-destined cells were decreased in the absence of VGF. The reduction of layer 4 cells in *Vgf*⁻/⁻ mice was not detected at P0 (*Figure 7—figure supplement 1C–F*, 14 sections from four wild-type and 15 sections from 4 *Vgf*⁻/⁻ mice for RORβ, 14 sections from four wild-type and 16 sections from 4 *Vgf*⁻/⁻ mice for EdU), indicating that layer 4-destined cells were decreased during postnatal period in *Vgf*⁻/⁻ mice. Next, cell fate of the EdU-labeled layer 4-destined cells was analyzed by immunohistochemistry at P7 (*Figure 7—figure supplement 1G*). The proportion of RORC-immunoreactive cells among the remaining EdU-positive cells in *Vgf*⁻/⁻ mice was reduced compared with wild-type, and that of cells expressing Brn2 (layer 2/3) and Ctip2 (layer 5) was not markedly increased (*Figure 7—figure supplement 1H*, 16 sections from four wild-type and *Vgf*⁻/⁻ mice for RORβ, 16 sections from four wild-type and *Vgf*⁻/⁻ mice for Brn2, 12 sections from four wild-type and *Vgf*⁻/⁻ mice for Ctip2). These results suggest that the number of, but not the fate of layer 4-destined cells was changed in *Vgf*⁻/⁻ mice during postnatal period. In contrast to *Vgf*⁻/⁻ mice, *Nrn1*⁻/⁻ mice exhibited a normal layer 4, such that the neuronal number was comparable to that in control mice (*Figure 7—figure supplement 2*, seven sections from four mice). Taken together, these results indicated that VGF regulates the neuronal number in layer 4 of S1, and that NRN1 is dispensable for this process.

To validate the significance of this regulation of layer 4 cell number in the further development of S1, the barrel formation, a unique feature observed in S1 in rodent (*Kawasaki, 2015*), was examined. While 5HTT-positive TCA terminals were distributed in a barrel-like manner (*Figure 6B*), the characteristic columnar repetitive arrangement of cells in layer 4 was impaired in *Vgf*-KO, such that cells were more evenly distributed along the tangential plane of layers in the mutant S1 revealed by a quantitative image analysis (*Figure 7J, K*, wild-type, four sections from four mice; *Vgf*⁻/⁻, five sections from five mice). Consistent with the presence of TCA terminals in *Vgf*-KO layer 4 (see *Figure 6B*), expression of *Btbd3* in layer 4 neurons, which is known to be dependent on neuronal activities of TCAs (*Matsui et al., 2013*), was substantially recognized unlike the TCA-ablated cases (*Figure 7—figure supplement 1I, J*, six sections from three mice). Therefore, the abnormal barrel organization in *Vgf*-KO is unlikely due to failure of activity-dependent processes, suggesting that the proper neuronal number in layer 4 is prerequisite for this process to operate. Taken together, these results suggested that VGF is essential for maintenance of layer 4 neuronal number and further formation of barrel structure in S1.

## Discussion

How the laminar configuration is formed in the developing cortex is one of the fundamental questions in developmental neurobiology, and the underlying molecular mechanisms have not yet been fully elucidated. Here, we investigated the influence of TCAs on cortical laminar formation by ablating TCAs from the thalamic VB in vivo. We found that the number of layer 4 neurons in S1 was decreased in the absence of TCAs during postnatal stages. This was rescued by overexpression of axon-derived secreted protein VGF in cortical layer 4. Furthermore, genetic disruption of *Vgf* resulted in a reduction of the layer 4 neuronal number. Collectively, these results indicated that TCAs are required for the formation of the area-specific laminar structure by regulating the number of layer 4 neurons through VGF. Our finding that TCA ablation reduced the number of layer 4 neurons by 33%, resulting in poorly distinguishable S1 from neighboring areas, indicates that TCAs play a substantial and major role in specialization of S1 characterized by a thick layer 4. Thus, highly site-specific axon projection, which depends on initial regional patterning of the target fields, in turn contributes to generation of further regional diversity and complexity within the target tissues in a remote fashion.

Our result that RORβ expression was reduced in the absence of TCAs is consistent with the previous studies on *Gbx2*-KO and *Celsr3/Dlx* mutant embryos, in which TCAs fail to be formed

(*Miyashita-Lin et al., 1999*; *Zhou et al., 2010*). However, we did not observe rewiring of TCAs regarding S1 as reported previously (*Pouchelon et al., 2014*; *Mezzera and López-Bendito, 2016*), although rewiring from the PO to S1 may have occurred to some extent as the PO was labeled more in several TCA-ablated specimens (see *Figure 1—figure supplement 3B*). This apparent inconsistency is perhaps due to the timing of TCA elimination. Although we did not compare the exact time course of TCA elimination among these different methods with our own hands, the method we employed provokes an acute ablation of TCAs upon DT administration at postnatal stages (see *Figure 2—figure supplement 2D*), which is beneficial for the aim of the present study specifically. Likewise, while it was reported that TCAs affect cortical neurogenesis (*Dehay et al., 2001*; *Gerstmann et al., 2015*; *Kraushar et al., 2015*), we have not found an obvious neurogenic abnormality in the TCA-ablated cortices. This could also be attributed to the timing of TCA elimination; TCAs may play the neurogenic role during the earlier stages, possibly upon their arrival at cortex since E15, which coincides with the peak of upper layer neuron production (*López-Bendito and Molnár, 2003*). Regarding this, we did not detect an obvious neurogenic phenotype in *Vgf*-KO mice at both P0 (*Figure 7—figure supplement 1D and F*) and P8 (*Figure 7G*), suggesting that VGF does not play a critical role in cortical neurogenesis.

While the present findings explain why layer 4 is thick in the somatosensory cortex, it is currently unclear whether the same mechanism accounts for the very thin layer 4 in the motor cortex. It would be interesting to test experimentally whether abnormal projection of TCAs to the motor cortex or artificial supply of TCA-derived factors in the motor cortex would thicken layer 4 and increase the neuronal number. While TCA-ablated mice showed almost no defect on V1, likely due to less cell death in dLGN which innervates V1, *Vgf*-KO mice exhibited significant reduction in layer 4 neurons in V1. These observations suggest that the regulation of the neuronal number of layer 4 by TCAs via VGF is a common mechanism operating widely in sensory areas.

The exact cellular events that led to the reduction in layer 4 neurons in the TCA-ablated cortex remain to be determined. Although most likely, specific cell death underlies this phenomenon, we could not detect obvious signs of cell death in this process. Previous studies have analyzed cell death in the postnatal cortex in rodents (*Ferrer et al., 1992*; *Gohlke et al., 2004*; *Spreafico et al., 1995*; *Thomaidou et al., 1997*; *Verney et al., 2000*), however, cell death in specific layers or areas in relation to the area-specific laminar structure has not been reported. Instead, a relatively uniform distribution of dying cells across layers and areas was reported in these studies. We also examined *Bax* and *Casp3* mutant mice; the laminar configuration across the areas appeared to be normal (HS, KS, unpublished observation). These observations suggest that relatively thinner layer 4 in other areas compared to S1 may not depend on specific cell death, yet it is still possible that it plays a role in the absence of TCAs in S1.

Another possibility for the reduction of layer 4 cells is neuronal migration. It has been reported that mismigration of layer 4-destined neurons to layers 2/3 upon knockdown of *Protocadherin 20* resulted in respecification of these neurons to acquire layer 2/3 characteristics, suggesting a positional influence on cortical neuron fate (*Oishi et al., 2016*). However, we found no evidence that neurons destined to layer 4 migrated to other layers and changed their fate in the TCA-ablated S1 (see *Figure 2E*, *Figure 2—figure supplement 2A*). Beside radial migration, neurons in layer 4 might migrate tangentially across areas, resulting in the area-specific laminar features in normal development. Such a process might have been impaired by the loss of TCAs, leading to a failure of accumulation of neurons into layer 4 of S1. To test this possibility, we tracked layer 4 neurons in the frontal cortex using the photo-switchable fluorescent protein kikGR (*Nishiyama et al., 2012*) in living pups from P0 to P7 by a laser-scanning confocal macroscopy. We did not detect any obvious signs of tangentially migrating layer 4 neurons; the labeled cells were retained within the original area of photoconversion in the motor area (HS and KS, unpublished results). Nevertheless, we might have overlooked cell migration because of limited detection sensitivity. In the future, it may be worth pursuing this possibility using more sophisticated and supersensitive methods recently published, through which the authors successfully observed activity-dependent dynamic dendrite rearrangement during the course of barrel formation in S1 (*Mizuno et al., 2014*). This method may allow us to observe if layer 4 neurons in S1 disperse into or fail to accumulate from surrounding areas in TCA-ablated mice. Similar technology might also enable us to determine whether layer 4 cells undergo cell death in the absence of TCAs or their actions through VGF.

There are two classical models for the development of cortical areas. One is the protomap model, in which the areal properties are predisposed in the regional identity of the cortical progenitors (*Rakic et al., 2009*). The other is the protocortex model, in which the cortical primordium is generated essentially homogeneous and is patterned into areas later by cues from the thalamic axons (*O'Leary, 1989*). In theory, these models are not mutually exclusive (*Sur and Rubenstein, 2005*), but can be reconciled as serial homology and refinement model recently proposed (*Cadwell et al., 2019*): cortical area development can be divided into serial three steps: (1) protomap, (2) area-specific maturation, and (3) activity-dependent refinement; processes postulated in the protocortex model are involved in (2) and (3). Previous studies on the roles of TCAs in cortical area development have led to the notion that TCAs indeed regulate areal size and characteristic gene expression in the cortex (*Chou et al., 2013*; *Pouchelon et al., 2014*; *Vue et al., 2013*; *Moreno-Juan et al., 2017*). Moreover, it has been reported that neural activity from TCAs controls neuronal morphology and barrel formation in a later stage (*Li et al., 2013*; *Narboux-Nême et al., 2012*). In this context, our present notion can be regarded as a process in (2) area-specific maturation: VGF released from TCA terminals controls the number of cortical neurons which is indispensable for the interaction between thalamic and layer 4 neurons. Since TCA terminals are sorted in a barrel-like manner in *Vgf*-KO S1 (*Figure 6B*), it is likely that the rearrangement or displacement of layer 4 neurons in response to TCAs, the second step of barrel formation (*López-Bendito and Molnár, 2003*), requires VGF.

Afferent-derived proteins play important roles in neural development. For example, ephrin A5 and Wnt3 from TCAs regulate differentiation of cortical neurons as mentioned above (*Gerstmann et al., 2015*; *Kraushar et al., 2015*). In other nervous systems, afferent-derived proteins regulate neurogenesis and synapse formation in their target region (*Huang and Kunes, 1998*; *Sanes and Lichtman, 2001*). In this study, we demonstrated that TCA-derived secretory protein VGF is necessary to maintain a sufficiently high neuronal number in cortical layer 4. VGF is widely expressed in the sensory thalamic nuclei, including the VB, dLGN, and MG (*Sato et al., 2012*), suggesting that this cue is commonly used for higher-order differentiation of sensory cortices. In fact, the number of layer 4 neurons in V1, which is a target of dLGN, was reduced in *Vgf*-KO mice (*Figure 7B and C*). Although disruption of NRN1, another TCA-derived secretory protein, did not affect the number of layer 4 neurons, it may play roles in other aspects of layer formation, for example, dendritic growth of spiny stellate neurons, as previously shown in vitro (*Sato et al., 2012*).

Our finding that a TCA-derived factor contributes to areal differentiation of the cerebral cortex may provide an insight into the generation of diversity in the areal pattern across mammalian species. In general, there is a strong correlation between dependence on particular sensory modality adopted for living environments and the proportional development of the corresponding areas (*Krubitzer, 2007*). In fact, experimental manipulation of the size of thalamic nuclei resulted in appropriate alteration of the areal pattern (*Chou et al., 2013*; *Vue et al., 2013*). Moreover, thalamic calcium wave plays critical roles in coordinating areal size prior to sensory processing (*Moreno-Juan et al., 2017*). Thus, mammals seem to have acquired coordinated evolution of thalamic sensory modality components and cortical areal characteristics to support their ecological diversity, resulting in the present prosperity worldwide. In this regard, it would be intriguing to explore the roles of the thalamus-derived factors, such as VGF or NRN1, along with the calcium wave mentioned above, in cortical area diversity among mammalian species. It is worth noting that both NRN1 and VGF are induced by neuronal activities in various brain regions including the cortex, thalamus, and hippocampus in later developmental stages and in adults (*Corriveau et al., 1999*; *Harwell et al., 2005*; *Snyder et al., 1997*). Although it is not known how these factors are induced in the thalamus during area formation, it could be a potential link between evolution of a peripheral sensory system and its corresponding cortical area.

## Materials and methods
### Animals
Time-pregnant ICR mice (SLC Japan) were used for immunohistochemistry, in utero electroporation and EdU administration. The day of vaginal plug detection was designated as embryonic day 0 (E0), and the day of birth as postnatal day 0 (P0). 5HTT-Cre (C57/BL/6J-*Tg(Slc6a4-cre)*208Ito) mice were previously generated (*Arakawa et al., 2014*). R26-EYFP (B6.129 × 1-*Gt(ROSA)26Sor*^tm1(EYFP)Cos^/J) mice were a generous gift from Dr. Constantini (Columbia University, USA) (*Srinivas et al., 2001*). R26-DTR

mice were purchased from The Jackson Laboratory (C57BL/6-*Gt(ROSA)26Sor*[tm1(HBEGF)Awai]/J). For cell ablation, 40 ng of DT was administrated once into pups of 5HTT-Cre; R26-DTR mice by intraperitoneal injection through the back skin at P0 (Calbiochem, #322326). Embryos and newborn pups of those transgenic, 5HTT-Cre; R26-DTR, *Vgf*-KO, and *Nrn1*-KO lines were obtained by conventional crossing as well as through a reproductive method established by the Center for Animal Resources and Development (CARD, Kumamoto University) to efficiently obtain a sufficient number of mice (*Takeo and Nakagata, 2015*). All experiments were carried out following the Guidelines for Laboratory Animals of Kumamoto University and the Japan Neuroscience Society.

## Plasmids

For forced expression of NRN1 and VGF, pCAG-*Nrn1*-Flag and pCAG-*Vgf*-Fc, respectively, were used (*Sato et al., 2012*).

For electroporation-based cell ablation, pCAG-*DTR* was constructed as follows: a cDNA fragment containing the coding region of human *HBEGF* (GenBank accession number: BC033097) was obtained from a commercial supplier (clone ID 100067676, DNAFORM). The coding region was first cloned into a pGEM-T Eeasy vector cut with HincII and EcoRV followed by adenine addition and ligation. The coding region was subcloned into the pCAG vector by digesting pGEM-T Easy-*DTR* and pCAG with EcoRI and ligating them. pCAG-*DsRed* was coelectroporated with these plasmids to monitor the electroporated regions in collected brains (*Zhao et al., 2011*).

For in situ hybridization of *Btbd3*, pFLCI-*Btbd3* (a generous gift from Dr. Tomomi Shimogori, RIKEN CBS, Japan) was used.

## Staining

Immunohistochemistry was conducted according to a standard protocol. Briefly, postnatal mouse brains were dissected and fixed with 4% paraformaldehyde (PFA) in phosphate-buffered saline (PBS) at room temperature (RT) for 3 hr and then incubated sequentially with 12.5% and 25% (wt/vol) sucrose-containing PBS (pH 7.4), sequentially. After the brains were frozen at −80°C, coronal sections of 20 or 40 μm were cut with a cryostat and collected in PBS containing 0.1% sodium azide. For staining with anti-RORC and -NRN1 antibodies, sections were mounted on slide glasses and treated with 10 mM sodium citrate buffer (pH 6.0) at 105°C for 10 min or antigen retrieval solution (HistoVT One, Nacalai Tesque) at 70°C for 20 min, respectively. For staining with anti-NeuN and anti-FLAG antibody, M.O.M. Basic Kit (#BMK-2202, Vector) was used to block detection of endogenous IgG. After blocking with PBS containing 5% normal goat/donkey serum and 0.1% Triton X-100 (blocking buffer) at RT for 1 hr, the sections were incubated at 4°C overnight with the following antibodies in the blocking buffer: mouse anti-RORC (1:800; catalog no. PP-H3925-00, Perseus Proteomics; although this antibody recognizes RORα, β and γ, RORγ is not expressed in the postnatal brain, and RORα expression generally overlaps with RORβ but is weak in the cortex [in Allen Brain Atlas]), rabbit anti-GFP (1:800; #A6455, Invitrogen), rabbit anti-Iba1 (1:500; catalog no. 019-19741, Wako), rabbit anti-5HTT (1:10,000; catalog no. 24330, ImmunoStar), goat anti-Brn2 (1:50; catalog no. sc-6029, Santa Cruz), rat anti-Ctip2 (1:200; catalog no. ab18465, Abcam), rabbit anti-Tbr1 (1:500; catalog no. ab31940, Abcam), rabbit anti-RFP (1:1000; catalog no. PM005, MBL), mouse anti-FLAG (1:1000, catalog no. F1804, Sigma-Aldrich), rabbit anti-ssDNA (1:300; catalog no. 18731, MBL), rabbit anti-RORβ (1:5000; catalog no. pAb-RORβHS-100, Diagenode), mouse anti-NeuN (1:400; catalog no. MAB377, Chemicon), rabbit anti-Cux1 (1:100; catalog no. sc-13024, Santa Cruz), rabbit anti-NRN1 (1:50; catalog no. sc-25261, Santa Cruz), and goat anti-VGF (1:50; catalog no. sc-10381, Santa Cruz). After three washes with PBS containing 0.1% Triton X-100 (PBST), the sections were incubated with Alexa 488- or Alexa 594-conjugated secondary antibody (1:500; Thermo Fisher Scientific) at RT for 2 hr. For nonfluorescent detection, sections were incubated with biotinylated secondary antibodies and processed using the ABC histochemical method (VECTASTAIN ABC Kit, Vector). After three washes with PBST, the sections were counterstained with a DAPI solution (1:1000, Wako, Japan) and embedded with a mounting agent (SlowFade Gold antifade reagent, Thermo Fisher Scientific).

For EdU labeling of layer 4 neurons and postnatally generated cells, 5 mg/ml EdU in PBS was intraperitoneally injected into pregnant mutant mice (50 μg/g) at E14.7 (for a normally mated mother) or E14.3 (for a mother with transplanted embryos), and into perinatal ICR mice once a day for 5 days

(50 µg/g), respectively. Frozen sections prepared as described above were stained for EdU using a detection kit (Click-iT EdU Imaging Kit, Thermo Fisher Scientific).

## In situ hybridization

In situ hybridization was performed as described previously (*Zhong et al., 2004*). DIG-labeled RNA probes were used for hybridization. RNA probes were synthesized from pFLCI-*Btbd3*. To produce linearized templates for the synthesis of riboprobes, plasmid was digested with NcoI. Linearized DNA was purified, and in vitro transcription was carried out (DIG-RNA Synthesis Kit; Roche). Finally, the probes were purified and kept at −80°C. Frozen sections were prepared in the same manner as immunohistochemistry described above. Sections were refixed in 4% PFA in PBS, washed three times in PBS. Prehybridization was carried out at 65°C for 1 hr in hybridization buffer (50% formamide, 5× SSC (saline-sodium citrate), 1% sodium dodecyl sulfate [SDS], 50 µg/ml of heparin, 50 µg/ml of tRNA), followed by hybridization overnight at 65°C in the same buffer containing 2 µg/ml of DIG-labeled RNA probe. After three washes in RNA wash buffer (50% formamide, 5× SSC and 1% SDS) at 60°C, the sections were blocked (blocking regent; Roche) for 1 hr at RT, and then incubated overnight at 4°C with alkaline phosphatase-conjugated anti-DIG antibody (1:1000; Roche). After five washes at RT, the color reaction was carried out at RT or 4°C in NBT/BCIP (Roche) in TBS (Tris-buffered saline). The reaction was terminated by immersing the sections in TE buffer (10 mM Tris–HCl, 1 mM EDTA [ethylenediamine tetraacetic acid], pH 8.0) for 10 min, and the sections were then fixed in 4% PFA in PBS for 15 min. The sections were treated in 70%, 80%, 90%, and 100% ethanol and xylene and then embedded.

## In utero electroporation

In utero electroporation was carried out as described previously (*Matsui et al., 2011*; *Saito and Nakatsuji, 2001*; *Tabata and Nakajima, 2001*) with slight modifications. To exogenously express *Nrn1* and *Vgf* in cortical layer 4, time-pregnant mutant mice of E14.3 were deeply anesthetized with pentobarbital (50 mg/kg). After the abdomen had been cleaned with 70% ethanol, a 3 cm midline laparotomy was performed, and the uterus was exposed. For DNA microinjection, plasmid DNA purified with the Plasmid Maxi-prep Kit (Genopure Plasmid Maxi Kit, Roche) was dissolved in Tris–EDTA buffer. Fast Green solution (0.1%) was added to the plasmid solution at 1:20 (vol/vol) ratio to monitor the injection. Approximately 1–2 µl of a mixture of 3–5 mg/ml pCAG-*Nrn1*-Flag and/or pCAG-*Vgf*-Fc and 1 mg/ml pCAG-*DsRed* was injected into the lateral ventricle with a glass micropipette. The embryos in the uterus were placed in a tweezers-type electrode equipped with two platinum discs of 5 mm in diameter at the tip (CUY650-5, Nepa Gene, Japan). Electronic pulses (30 V, 50 ms) were delivered four times at intervals of 950 ms with an electroporator (NEPA21, Nepa Gene, Japan), and then, the uterine horns were placed back into the abdominal cavity. The abdominal wall and skin were sewed up with surgical sutures, and the embryos were allowed to develop until P7. For electroporation-based cell ablation of thalamic neurons, time-pregnant ICR mice of E11.5 were used and a mixture of 3 mg/ml pCAG-*DTR* and 1 mg/ml pCAG-*DsRed* was injected into the third ventricle, followed by application of electric pulses (25 V, 50 ms).

## Generation of *Nrn1* and *Vgf*-deficient mice by the CRISPR/Cas9 method

To delete *Nrn1* and *Vgf* loci, one and two sgRNAs targeting the *Nrn1* and *Vgf* exons, respectively, were designed (*Nrn1* sgRNA: AGCATGGCCAACTACCCGCA; *Vgf* sgRNA: TCACGTTGCCGGCATC CGTC, CGGTACTGTTGCAGGCACTGGACCGT). Fertilized eggs derived from C57BL/6J mice were electroporated with a mixture of sgRNAs and Cas9 protein and transplanted into foster mother mice. Brains were collected from pups at P8 and fixed with 4% PFA in PBS for at RT 3 hr. Before fixation, a piece of cerebellum was dissected. Genomic DNA was extracted from it and Nrn1 and Vgf loci were analyzed by sequencing. For genomic sequencing, DNA fragments including the sgRNA sequences were amplified by PCR using the following primers: *Nrn1*: 5′-ACCAGGGAACTGAGCC TGAG-3′ and 5′-GGACTCACCTCCCTGCTATC-3′; *Vgf*: 5′-GGTACCCAGAAGGAGGATTG-3′ and 5′-TTGCTCGGACTGAAATCTCG-3′. Sequencing PCR was performed using the amplified DNA fragments as a template and the primers: *Nrn1*: 5′-ACCAGGGAACTGAGCCTGAG-3′; *Vgf*: 5′-GGTA CCCAGAAGGAGGATTG-3′ (near sgRNA#1) or 5′-CTCAGCTCTGAGCATAATGG-3′ (near sgRNA#2).

Mice harboring a deletion that causes frame shift resulting in translation failure without wild-type sequences nor deletion in multiples of three bases resulting in truncated protein product were designated null mutants.

## Lipophilic dye labeling

For labeling neural connections, P7 mutant mouse brains were fixed with 4% PFA in PBS at RT for 3 hr. A small crystal of DiA and DiI (catalog no. D3883 for DiA and D3911 for DiI, Thermo Fisher Scientific) was inserted into S1 and V1, respectively. Incubated in 4% PFA in PBS at 37°C for 2 weeks after implantation, the brains were cut into 100 μm slices with a vibratome (Leica) and observed by fluorescence microscopy (BX52, Olympus).

## Data quantification and statistical analysis

Marker-positive or EdU-labeled cells were counted in photomicrographs of single focal plane (850 × 850 μm) acquired with a laser scanning confocal microscope (LSM780, Zeiss) with a 10× objective lens. After sections with inappropriate histology or staining were excluded, particles of >13.37 μm² with signal intensities higher than a given threshold were quantified using a Metamorph software (Molecular Devices). The relative cell number as a percentage of control was determined for each section, then the average value of the collection of sections for each experimental case was calculated. To measure the thickness of the marker-positive layer, the maximum radial distance of the tangential lines, along which more than two positive particles defined as above were distributed, was measured automatically by Metamorph. The relative thickness and standard error of the mean were calculated for each experimental condition. To count the number of ssDNA-positive cells, 995-μm-wide strip of the upper (2–4) and deep (5–6) layers and VB region in photomicrographs acquired with a fluorescent microscope (BX52, Olympus) with a ×4 objective lens were extracted. Particles of >1.49 μm² with signal intensities higher than a given threshold, were quantified using Metamorph software. To measure the distribution variation of layer 4 cells in the barrel field, the region of 25 μm in height, and 583 μm in width, was extracted from confocal images of the barrel field stained with DAPI. The region was subdivided into 35 (25 × 16.7 μm each) and total intensity was measured for each subdivision with ImageJ. The variation of intensities among 35 subdivisions was calculated for each section. Group means were compared using Mann–Whitney $U$ test with GraphPad PRISM (GraphPad Software), and *$p < 0.05$ was regarded statistically significant.

## Acknowledgements

The authors thank J Kusuura for excellent technical support. We are also grateful to Drs T Shimogori and Y Yamaguchi for discussion, reagents, and mutant mice. This work was supported by the Liaison Laboratory Research Promotion Center at IMEG and the Reproductive Technology Team of CARD Mouse Bank at CARD Kumamoto University.

## Additional information

### Funding

| Funder | Grant reference number | Author |
|---|---|---|
| Japan Society for the Promotion of Science | KM101-2587054400 | Haruka Sato |
| Japan Society for the Promotion of Science | KM100-2633200 | Haruka Sato |
| Japan Society for the Promotion of Science | KM101-18K1483900 | Haruka Sato |
| Ministry of Education, Culture, Sports, Science and Technology | JP06J08049 | Jun Hatakeyama |

| Funder | Grant reference number | Author |
| --- | --- | --- |
| Ministry of Education, Culture, Sports, Science and Technology | JP21870030 | Jun Hatakeyama |
| Ministry of Education, Culture, Sports, Science and Technology | JP24790288 | Jun Hatakeyama |
| Ministry of Education, Culture, Sports, Science and Technology | JP15K19011 | Jun Hatakeyama |
| Ministry of Education, Culture, Sports, Science and Technology | JP16H01449 | Jun Hatakeyama |
| Ministry of Education, Culture, Sports, Science and Technology | JP17H05771 | Jun Hatakeyama |
| Ministry of Education, Culture, Sports, Science and Technology | JP16H06276 | Kimi Araki |
| Ministry of Education, Culture, Sports, Science and Technology | 18GS0329-01 | Kenji Shimamura |
| Ministry of Education, Culture, Sports, Science and Technology | JP16K07375 | Kenji Shimamura |

The funders had no role in study design, data collection, and interpretation, or the decision to submit the work for publication.

## Author contributions

Haruka Sato, Conceptualization, Data curation, Formal analysis, Funding acquisition, Investigation, Methodology, Validation, Writing - original draft; Jun Hatakeyama, Formal analysis, Funding acquisition; Takuji Iwasato, Resources; Kimi Araki, Funding acquisition, Methodology, Resources; Nobuhiko Yamamoto, Conceptualization, Resources; Kenji Shimamura, Conceptualization, Funding acquisition, Investigation, Project administration, Supervision, Validation, Writing - review and editing

## Author ORCIDs

Haruka Sato http://orcid.org/0000-0001-6839-0146
Kenji Shimamura http://orcid.org/0000-0001-7102-6513

## Ethics

This study was performed in strict accordance with the guidelines for laboratory animals of Kumamoto University and the Japan Neuroscience Society. All of the animals were handled according to approved institutional animal care and protocols by the Committee on the Ethics of Animal Experiments of Kumamoto University (Permit Numbers: 27-124, A29-080, 2019-110, 2020-055). All surgery was performed under sodium pentobarbital anesthesia, and every effort was made to minimize suffering.

## Decision letter and Author response

Decision letter https://doi.org/10.7554/eLife.67549.sa1
Author response https://doi.org/10.7554/eLife.67549.sa2

# Additional files

## Supplementary files

• Transparent reporting form

## Data availability

All data generated or analysed during this study are included in the manuscript and supporting files; Source data files have been provided for Figures 1-7 and Figure 1-figure supplement 1, Figure 2-figure supplement 1, 2, Figure 5-figure supplment 2, Figure 7-figure supplement 1, 2.

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

## Appendix 1

### Appendix 1—key resources table

| Reagent type (species) or resource | Designation | Source or reference | Identifiers | Additional information |
|---|---|---|---|---|
| Gene (*M. musculus*) | *Nrn1* | GenBank | BC035531 | |
| Gene (*M. musculus*) | *Vgf* | GenBank | BC085134 | |
| Gene (*Homo-sapiens*) | *HBEGF* | GenBank | BC033097 | |
| Strain, strain background (*Escherichia coli*) | DH5α | Our lab | | Calcium chloride-treated competent cells |
| Strain, strain background (*M. musculus*) | ICR | SLC Japan | Slc:ICR | |
| Strain, strain background (*M. musculus*) | C57BL/6J | SLC Japan | C57BL/6JJmsSlc | |
| Genetic reagent (*M. musculus*) | C57BL/6J-Tg(Slc6a4-cre)208Ito (5HTT-Cre, Sert-Cre, SLC6A4-Cre) | RIKEN BRC **Arakawa et al., 2014** | RBRC10598 C57BL/6J-*Tg(Slc6a4-cre)208Ito* | |
| Genetic reagent (*M. musculus*) | ROSA26iDTR (B6-iDTR) | The Jackson Laboratory **Buch et al., 2005** | Strain #007900C57BL/6-Gt(ROSA)26Sor$^{tm1(HBEGF)Awai}$/J | |
| Genetic reagent (*M. musculus*) | R26R-EYFP | The Jackson Laboratory **Srinivas et al., 2001** | Strain #006148 B6.129 × 1-*Gt(ROSA)26Sor$^{tm1(EYFP)Cos}$*/J | |
| Genetic reagent (*M. musculus*) | *Nrn1*-KO | This paper | | CRISPR-mediated *Nrn1* null mice available upon request |
| Genetic reagent (*M. musculus*) | *Vgf*-KO | This paper | | CRISPR-mediated *Vgf* null mice available upon request |
| Antibody | anti-RORC (Mouse monoclonal) | Perseus Proteomics | Cat#: PP-H3925-00 | IF (1:800) |
| Antibody | anti-GFP (Rabbit polyclonal) | Invitrogen | Cat#: A6455 | IF (1:800) |
| Antibody | anti-Iba1 (Rabbit polyclonal) | Wako | Cat#: 019-19741 | IF (1:500) |
| Antibody | anti-Brn2 (Goat polyclonal) | Santa Cruz | Cat#: sc-6029 | IF (1:50) |
| Antibody | anti-Ctip2 (Rat polyclonal) | Abcam | Cat#: ab18465 | IF (1:200) |
| Antibody | anti-Tbr1 (Rabbit polyclonal) | Abcam | Cat#: ab31940 | IF (1:500) |
| Antibody | anti-NeuN (Mouse monoclonal) | Chemicon | Cat#: MAB377 | IF (1:400) |
| Antibody | anti- Cux1 (Rabbit polyclonal) | Santa Cruz | Cat#: sc-13024 | IF (1:100) |
| Antibody | anti-RFP (Rabbit polyclonal) | MBL | Cat#: PM005 | IF (1:1000) |
| Antibody | anti-FLAG (Mouse monoclonal) | Sigma-Aldrich | Cat#: F1804 | IF (1:1000) |
| Antibody | anti-ssDNA (Rabbit polyclonal) | MBL | Cat#: 18,731 | IF (1:300) |
| Antibody | anti- RORβ (Rabbit polyclonal) | Diagenode | Cat#: pAb-RORβHS-100 | IF (1:5000) |
| Antibody | anti- NRN1 (Rabbit polyclonal) | Santa Cruz | Cat#: sc-13 25,261 | IF (1:50) |

*Appendix 1 Continued on next page*

*Appendix 1 Continued*

| Reagent type (species) or resource | Designation | Source or reference | Identifiers | Additional information |
|---|---|---|---|---|
| Antibody | anti-VGF (Goat polyclonal) | Santa Cruz | Cat#: sc-10381 | IF (1:50) |
| Antibody | anti-DIG (Sheep polyclonal) | Roche | Cat#: 11093274910 | ISH (1:1000) |
| Recombinant DNA reagent | *pFLCI-Btbd3* (plasmid) | *Matsui et al., 2011* | | |
| Recombinant DNA reagent | *pCAG-DTR* (plasmid) | This paper | | *HBEGF* expression plasmid driven by CAG promoter |
| Recombinant DNA reagent | *pCAG-DsRed* (plasmid) | *Zhao et al., 2011* | | |
| Sequence-based reagent | *Nrn1* sgRNA | This paper | Single-guide RNA targeting *Nrn1* | AGCATGGCCAACTACCCGCA |
| Sequence-based reagent | *Vgf* sgRNA #1 | This paper | Single-guide RNA targeting *Vgf* | TCACGTTGCCGGCATCCGTC |
| Sequence-based reagent | *Vgf* sgRNA #2 | This paper | Single-guide RNA targeting *Vgf* | CGGTACTGTTGCAGGCACTG GACCGT |
| Sequence-based reagent | *Nrn1_FW* | This paper | PCR primer | ACCAGGGAACTGAGCCTGAG |
| Sequence-based reagent | *Nrn1_RV* | This paper | PCR primer | GGACTCACCTCCCTGCTATC |
| Sequence-based reagent | *Vgf-FW* | This paper | PCR primer | GGTACCCAGAAGGAGGATTG |
| Sequence-based reagent | *Vgf-RV* | This paper | PCR primer | TTGCTCGGACTGAAATCTCG |
| Sequence-based reagent | *Vgf-Seq* | This paper | Sequencing primer | CTCAGCTCTGAGCATAATGG |
| Peptide, recombinant protein | Diphtheria toxin | Calbiochem | Cat#: 322,326 | |
| Commercial assay or kit | HistoVT One | Nacalai Tesque | Cat#: 06380-05 | |
| Commercial assay or kit | M.O.M. Basic Kit | Vector | Cat#: BMK-2202 | |
| Commercial assay or kit | VENTASTAIN ABC Kit | Vector | Cat#: PK4000 | |
| Commercial assay or kit | Click-iT EdU Imaging Kit | Thermo Fisher Scientific | Cat#: C10337 | |
| Commercial assay or kit | DIG-RNA Synthesis Kit | Roche | Cat#: 11175025910 | |
| Commercial assay or kit | Plasmid Maxi Kit, Roche | Genopure | Cat#: 03143422001 | |
| Chemical compound, drug | NBT/BCIP | Roche | Cat#: 11681451001 | ISH (1:50) |
| Chemical compound, drug | DiA | Thermo Fisher Scientific | Cat#: D3883 | |
| Chemical compound, drug | DiI | Thermo Fisher Scientific | Cat#: D3911 | |
| Software, algorithm | Metamorph software | Molecular Devices | | |
| Software, algorithm | ImageJ | https://imagej.nih.gov/ij/ | ImageJ v1.52 | |
| Software, algorithm | GraphPad Prism | GraphPad Software | GraphPad Prism 5.0 | |
| Other | DAPI stain | Wako | Cat#: 340-07971 | 1:1000 |

