## [Editor Report]

Here Sato and colleagues in the Shimamura lab investigate the role of extrinsic factors in the development of the murine neocortex. They show that factors provided by thalamocortical afferents, namely, VGF, are instructive in the formation of layer 4 neuron populations and thus layer thickness, particularly for the primary somatosensory cortex. This study in vivo, based on VGF knockouts, expands their previous findings in vitro on this process, and adds to our understanding of how cytoarchitectonic differences across cortical areas are established.

---

## [Decision Letter]

**Decision letter after peer review:**

[Editors’ note: the authors submitted for reconsideration following the decision after peer review. What follows is the decision letter after the first round of review.]

Thank you for submitting your work entitled "Thalamocortical axons control the cytoarchitecture of neocortical layers by area–specific supply of secretory proteins" for consideration by *eLife*. Your article has been reviewed by 3 peer reviewers, and the evaluation has been overseen by a Reviewing Editor and a Senior Editor. The following individual involved in review of your submission has agreed to reveal their identity: Zoltan Molnar (Reviewer #3).

We are sorry to say that, after consultation with the reviewers, we have decided that your work will not be considered further for publication by *eLife*.

Your study on the role of factors secreted by thalamocortical axons during the establishment of cortical layers of the murine neocortex was viewed with great interest to the editors and reviewers, and is a welcome segue from your previous work that Vgf and Nrn1 deriving from thalamocortical neurons contribute specifically to layer 4 neuron development in vitro. By postnatally ablating thalamocortical projections and studying a Vgf knockout, you further investigate the role of these projections in layer 4 establishment and bring some light to the in vivo role of VGF. As you can see from the Public Reviews, the reviewers found your study to be relevant and important to the question of the role factors derived from the thalamic afferents themselves, after intrinsic factors are at play, in cortical layer development. Nonetheless, they have numerous suggestions in the Recommendations for authors, also below, for amendments and further analyses, especially on the quantification, factor expression, assessment of cell death, and other areal and layer impairments in your in vivo genetic and experimental analyses.

Please note that we aim to publish articles with a single round of revision that under normal circumstances can be accomplished within two months. This means that work that has potential, but in our judgment would need extensive additional work, will not be considered for in–depth review. We do not intend any criticism of the quality of the data or the rigor of the science. We wish you good luck with your work and we hope you will consider *eLife* for future submissions.

*Reviewer #1:*

Sato et al. investigated the role of the thalamus–derived factor, VGF, as extrinsic cue that controls layer 4 development in the cortex. They show that this process is necessary for further maturation of the primary somatosensory cortex (S1) and the barrel field formation. To explore the role of thalamocortical axon (TCA) projections in cortical layer formation, the authors developed a mouse model with TCA ablation from the ventrobasal (VB) thalamic nucleus by the administration of diphtheria toxin (DT) from P0. They induced a decrease in the VB nucleus size and the TCAs terminals were also diminished in layer 4 of S1. Sato et al. demonstrated that the number of layer 4 cells in S1 is reduced in TCA–ablated model and to verify that these TCA ablation from the VB was indeed responsible for this laminar phenotype, they used a different approach to specifically ablate only the VB neurons. By performing a technically impressive in utero electroporation at e11.5 in the thalamus with a diphtheria toxin receptor (DTR) expression plasmid and then administered DT at P0, they mimicked the laminar phenotype in the cortex and the results were similar to the TCA–ablated mice. Moreover, they showed that, apparently, the rest of the cortical layers remain intact. Interestingly, the authors demonstrated that VGF and NRN1 as TCA–derived factors play an important role to maintain the layer 4 neuronal number during cortical development by restoring these cell number after the overexpression of these factors in TCA–ablated model in vivo. To further address this question, they induced a genetic inactivation of Vgf by using CRISPR/Cas9–mediated gene editing, and they proved that VGF is necessary for the maintenance of layer 4 neurons number.

The manuscript shows potentially interesting findings but there are some open questions and experiments that should be done to better support the conclusions of the paper. Moreover, some aspects of image acquisition and data analysis need to be clarified.

1) The figures need to be reorganized to achieve a better flow of the manuscript. For example, Figure 1 does not contribute to the story, and it would make more sense to transfer it to supplementary figures. Moreover, the Figure 5—figure supplement 1 could be a principal figure to show the absence of VGF in thalamus and cortex of TCA–ablated mice because this leads to wonder about the role of this gene and justify the following experiments of overexpression and knock–down.

2) In Figure 2—figure supplement 1 the authors show lipophilic dyes tracings assuming that they only labeled TCAs and thalamic neurons retrogradely. However, this is a conceptual mistake because these tracers label indistinctly, retrogradely and anterogradely. They inject the dyes at P7 when corticothalamic axons have already innervated the thalamus and thus presumably, the labelling shown in PO is, at least in part, from S1 layer 5. It would be better to use a L4 specific virus to trace only retrogradely projections. Moreover, they state that the number of labeled cells were similar in TCA–ablation and control mice, however, there is a need to increase the n to support this statement as there is a strong variability in this technique due to the change in labelling related the change in the size of the dyes.

3) There are some affirmations in the manuscript that would need a quantification for support. For example, when the authors state that dLGN was not reduced in TCA–ablated mice would need a quantification. The same in Figure 2A where it is stated that 5HTT–Cre is expressed in dLGN, and in Figure 2D, where the authors show a reduction in the size of the nucleus. Therefore, dLGN size quantifications should be done and it would be interesting to show a slice of 5HTT–Cre;R26–YEFP in V1 layer 4 showing 5HTT–Cre expression at P0–P7.

4) The results for cell death are not convincing, the authors should improve the detection method, for example, they could use caspase 3 in postnatal brains. In Figure 3—figure supplement 1 they only show quantifications in layer 2/3 and layer 4 where there is an increased tendency for cell death, but again the n is too small for some conditions. Additionally, in Figure 3—figure supplement 1C a quantification should be provided for cell death in layer 6. It is very likely that this cell death is due to the expression of Cre recombinase in layer 6 in the TCA–ablated mice. Thus, the results for cell death are not clear enough and some experiments should be repeated and /or conclusions clarify.

5) The authors sometimes underestimate the interpretation of the results. For example, in Figure 4 Tbr1 positive cells (layer 6) are reduced in TCA–ablated mice, and thus indicate that layer 6 is also affected in this mouse. Therefore, this model might not be the best one to investigate how extrinsic cues mechanisms regulate layer 4 development.

6) It would be interesting to explore the neural activity of layer 4 in the somatosensory cortex of Vgf–KO and TCA–ablated mice. Behavioral experiments with environmental enrichment and c–fos immunostaining would demonstrate the laminar functionality of the cortex under these different conditions.

*Reviewer #2:*

Sato H. and colleagues here investigate the role of extrinsic factors in the development of the murine neocortex. It was previously shown by these authors that thalamocortical neurons, through the expression of Vgf and Nrn1 among others, contribute specifically to layer 4 neurons development in vitro. In the current study, by postnatally ablating thalamocortical projections and studying a Vgf knockout, they further investigate the role of these projections in layer 4 establishment and bring some light to the in vivo role of VGF.

Although this is an interesting study, the novelty is relatively limited as it incrementally builds on previous work from this laboratory as well as previous work from several laboratories directly addressing the effects of ablation of specific thalamic nuclei on cortical neuron identity. In order to increase interest and relevance, several key experiments should be performed.

– It is interesting to see that RORC staining emerges in a rostrocaudal manner from P0 to P7. With the current data, the authors could perhaps assess whether this emergence also occurs in a mediolateral manner. This would help to better describe the emergence of the layer 4 across cortical areas. Moreover, to confirm the statement of layer 4 being thicker in S1, a quantification or a reference to quantification of layer 4 across cortical areas would be required.

– Some of the quantifications and experiments present only 2 cases. The authors should consider at least to increase the number of cases to 3. In addition, considering the number of cases in most of the experiments, the use of parametric statistical analyses is incorrect (not possible to assess dataset normality with less than 30 data points). The authors should switch to non–parametric statistical analyses.

– The use of a Cre–dependent DT strategy to ablate thalamic neurons is interesting to study the role of their projections postnatally. However, it seems important to better analyze the dynamics of their death after DT administration. The authors should try to assess, both thalamic and cortical, cell death from 24h post DT administration up to their analysis time point (P7). It is important to understand the dynamics of cell death in both regions to try to establish a clearer correlation. This could by assessed by performing Caspase3 immunolabelling at P1, P2, P3, P4, P5, P6, and P7 after P0 DT injection both in the thalamus and the cortex. In Figure 5C, it is unclear why there is no Dd–red signal in the control and DTA conditions.

– The authors should consider further describing the impact of this strategy on deep–layer neurons (YFP+ in Figure2A). As they are Cre+ during their developmental time course, DT is most likely going to target them (which could explain the reduced thickness of deep layers when compared to controls). This could be added as supplemental information, but it is important to evaluate the ground state of the neocortex to evaluate the real impact of TCAs.

– For the panels containing thalamic nuclei, it would help to have white dotted lines (as in Figure 2D) around them in all the panels. As an example, it is difficult to determine the MG position in DiA images (Figure 2—figure supplement 1).

– Throughout the paper, the fluctuation of layer 4 number of neurons throughout conditions is assessed by the expression of RORC (example: Figure 3C). In addition, to investigate possible layer 4 fate changes, the authors quantify L2/3 or L5/6 number of cells (with known markers). In order to improve the analysis of layer 4 neurons, instead of using postnatal markers, it would perhaps be interesting to electroporate at E14 to have permanent labelling of this cell population. On the one hand, it would help at first to assess if these specific cells are dying, and on the other, it would better allow the analysis of possible fate changes. This methodology would be complementary to the actual Cre–DTR system, making it optimal for this type of analysis. Another possibility would be to inject birth dating markers embryonically, such as CFSE.

– Regarding Nrn1 and Vgf rescue experiments, it is interesting to see that the absence of TCAs promotes their absence in layer 4. It would be interesting to add whether the over–expression of Nrn1 and Vgf, alone, can maintain "normal" levels of RorC+ cells in layer 4. Moreover, besides analyzing the presence of DsRed (Figure 5C) in double electroporations, it would be interesting to see the presence of the proteins themselves (either by ISH or immunolabelling) as a positive control for the experiment.

– Additionally, it would be interesting to visualize the expression of Vgf and Nrn1 receptors expression by layer 4 cells and their expression dynamics. Is it constantly expressed or there is a critical period for their expression? Can the authors rescue L4 neurons by overexpressing Vgf in L2/3 neurons? This would demonstrate a non–cell–autonomous effect onto L4 neurons.

– In Vgf–/– mutants, there is an increase of Brn2+ and Ctip2+ cells, which could correspond to a fate change of layer 4 cells in absence of Vgf. It would be interesting to electroporate Gfp at E14 to label layer 4 cells and analyze the expression of these markers at P7.

– For Nrn1–/– analysis of layer 4 cell number, there should be a quantification. It is hard to tell from the images alone if there is a difference or not.

*Reviewer #3:*

The first extrinsic influences that shape the cortical neuroepithelium are secretory factors, emanating from signaling centers adjacent to the telencephalic vesicles. These centers set up the gross areal pattern of the neocortex without any extrinsic signals. While it is clear that there are intrinsic gradients from the beginning of cortical neurogenesis, there are also extrinsic cues that contribute to the differences. The best candidate to deliver the area–specific cues to the cortex is via area–specific thalamocortical projections. These arrive to the cortex very early, at the peak of the cortical neurogenesis and neuronal migration. The impact of thalamic lesions on cortical lamination was demonstrated by Windrem and Finlay, 1991. Moreover, the influence of ephrin A5 on cortical progenitor cells and the effect of secreted Wnt3 on neuronal differentiation has been previously described.

The Sato et al., paper studies two thalamus–derived factors that might mediate some of the area–specific differences to the cortex. The authors used the results of their previous screens to identify secreted molecules that are not produced in the cortex, but delivered to the cortex through the thalamocortical projections. The authors screened for thalamus–specific genes by comparing expression profiles of the thalamus and the cortex (Sato et al., 2012). These screens identified neuritin 1 (NRN1) and VGF nerve growth factor inducible (Vgf) transported to the cortex through TCAs. This study demonstrates that VGF maintains the proper amount of layer 4 neurons in S1.

The study is reported in a logical sequence. First the authors established that birthdates of all cortical layers in various cortical areas are all prenatal. Since all neurons were generated before birth, no NeuN positive birthdated cells were found after postnatal EdU injection and therefore it is unlikely that layer 4 neurons are additively generated in S1. The study also analysed the emergence of RORβ (RAR–related orphan receptor β) expression in the postnatal primary somatosensory cortex.

Then, they performed toxin–mediated ventrobasal complex ablation in vivo. Cre expression was the highest in VB among the thalamic nuclei in 5HTT–Cre mice, therefore the lesion was the greatest there. This reduced the thalamocortical axons that project to primary somatosensory cortex. The ablation triggered accumulation of Iba1–immunoreactive microglial cells in VB. This indicated that these are the regions with dead cells, in the thalamus. After this ablation VB was reduced at P5, and RORα–expressing thalamic neurons were decreased.

The authors examined the possibility that other thalamic neurons project to the S1, but detailed tracing from S1 did not produce backlabeling pattern in the thalamus that would indicate TCA ingrowth from the dLGN or MG to S1. No such re–wiring was observed in the PO VB TCA–ablated mice.

The number of RORβ–expressing cortical cells was decreased to 67% as compared with control cortex. Moreover, the absolute number of layer 4 neurons also decreased in the TCA–ablated S1. This argued against the possibilities of altered RORβ expression or fate change of the layer 4–destined cells to those of other layers. The authors suggested cell death as a possible mechanism for getting these differences. However, all the conventional cell death detection methods (ssDNA, cleaved caspase 3, Iba1, mRNA of Bax, Bad, and Bak, and DAPI), they could not obtain convincing evidence for significant cell death induction in layer 4 upon TCA ablation.

The toxin–mediated ventrobasal complex ablation in vivo is based on cre expression. Since Cre expression was the highest in VB among the thalamic nuclei in 5HTT–Cre mice, this was the region for the greatest damage. Nevertheless, there was additional cre expression in layer 6 and also in the raphe nucleus the authors designed experiments to exclude that the layer 4 reduction in 5HTT–Cre; R26–DTR mice is due to ablation of these brain parts rather than the VB in the thalamus. The authors used DTR expression plasmid that was electroporated into the embryonic dorsal thalamus in utero at E11.5, when VB neurons are generated. The authors demonstrated that the effect was S1 specific. The cortical areas with dLGN and MGN innervation were not affected. Layers 2/3 (revealed with Brn2), layer 5 (revealed with Ctip2) and layer 6 (revealed with Tbr1) appear intact upon TCA elimination suggesting that the effect was specific for layer 4. The number of layer 4 neurons is restored by forced expression of NRN1 and VGF in the cortex of TCA–ablated mice NRN1 and VGF was lost in the thalamic nuclei and their axon terminals in layer 4 of S1 in TCA–ablated mice

The next group of experiments demonstrated that genetic inactivation of Vgf in thalamocortical projection neurons results in a reduction in layer 4 neurons in S1 cortex and that NRN1 is dispensable in this process. The authors used three single–guide RNAs (sgRNAs) cutting exons of Nrn1 and Vgf to induce frame shifts resulting in failure of protein translation of both NRN1 and VGF. They electroporated the sgRNAs and Cas9 protein into fertilized eggs, mutations were induced in the genomic sequences of Nrn1 and Vgf allele near designed sgRNAs. The loss of TCA–derived VGF from the cortex resulted in the significant reduction of RORβ immunoreactive layer 4 cells in S1 and in V1. These observations suggest that the regulation of the neuronal number of layer 4 by TCAs via VGF is a common mechanism operating widely in sensory areas.

VGF released from TCA terminals sets the exact numbers of cortical layer 4 neurons. Then, the activity dependent sorting of thalamocortical afferents will impose the cytoarchitectonic changes that will form the cytoarchitectonic barrels. This interaction between thalamic and layer 4 neurons to form the cytoarchitectonic barrel formation in S1 was significantly impaired in Vgf–KO mice despite the presence of TCAs. The paper convincingly demonstrates that thalamocortical axons play instructive roles in the regulation of layer 4 cell numbers and the specification of area properties of somatosensory and visual cortices.

I consider this study a very important and significant step in our field. This is a direct demonstration of the instructive role of the thalamocortical projections. The major conclusions of the study will have to be supported by additional experiments (cell death – fate change distinction).

Overall, this is an excellent study with important findings.

1. While the authors clearly observed that layer 4 exhibited marked reduction in the number of neurons specifically within the primary somatosensory cortex and they reported no change in the number of layers 2–3, 5 or 6 neurons, I would be still interested in the neuron numbers in the entire depth of the cortex in an arbitrary unit column. "The number of RORβ–expressing cortical cells was decreased to 67% as compared with control cortex. Moreover, the absolute number of layer 4 neurons also decreased in the TCA–ablated S1." This argued against the possibilities of altered RORβ expression or fate change of the layer 4–destined cells to those of another layer. Is the reduction being just because the effect to layer 4?

2. While the study is commenting on the changes within S1, it does not comment on the possibility that the size of S1 changed. It would be good to show the areal extension of the S1 barrel field on flatmounts and compare the representation of the S1 and V1 after all these manipulations. However, it is not essential for the basic conclusions of the study.

3. Formation of the barrels requires two stages. The periphery related sorting of the thalamocortical afferents (revealed with stains for thalamocortical afferents 5HTT or DiI or CO staining) and then, the activity dependent sorting of thalamocortical afferents will impose the cytoarchitectonic changes that will form the cytoarchitectonic barrels (revealed with DAPI, Nissl or NeuN staining) (see Lopez–Bendito and Molnar, 2003). It would be good to clarify which mechanism is affected.

4. The authors suggested cell death in layer 4 as a possible mechanisms for getting these differences. However, all the conventional cell death detection methods (ssDNA, cleaved caspase 3, Iba1, mRNA of Bax, Bad, and Bak, and DAPI), they could not obtain convincing evidence for significant cell death induction in layer 4 upon TCA ablation. Was there any change in microglia distribution in layer 4 after the ablation of the VB thalamic neurons?

Even though the authors justify a possible technical difficulty in detecting dead cells in a dynamic developing system (which I completely agree with and understand), they don't provide a solid explanation for such a great decrease in cell number in layer 4. As the authors describe layer 4 neurons are already generated at the time of ablation, therefore they either die or change their fate. The cell fate change should be better investigated and discussed. Can the loss of 67% of neurons be completely due to cell death?

5. There are numerous studies that demonstrated that six layered cortex can develop relatively normally without thalamic connections (Miyashita–Lin et al., 1999 DOI: 10.1126/science.285.5429.906; Zhou et al., 2010, DOI: https://doi.org/10.1523/JNEUROSCI.6005–09.2010). Did any of these studies suggested reduced layer 4 cell numbers? Do the authors used more sensitive methods for their detection? These issues should be discussed in more detail.

6. The study described that after ablation at P0, VB was significantly reduced by P5. Parallel with this RORα–expressing thalamic neurons were also decreased. The traced from S1 and did not see any signs of TCA rewiring from the dLGN or MG to S1. Why do the authors think that Mezzera and López–Bendito, (2016) had different findings? Is is because of the methods, timing, tracing?

7. The results explain why layer 4 is thick in the somatosensory cortex. However, it is not known whether the same mechanism operates in the motor cortex where there is a very thin layer 4. The authors reduce the possibility of a cell death–based mechanism by examining the Bax and Casp3 mutant mice. In this lone the laminar configuration across the areas appeared to be normal.

Do the authors suggest that the thinner layer 4 in M1 is the result of a different neurogenetic program generating these proportions?

8. Do the authors suspect possible interaction with neurogenesis if the VB is ablated from early embryonic stages? I am not requesting this experiment, just some speculations in the discussion.

In fact, TCA ablation at earlier periods of development is expected to have a greater effect on neurogenesis, as the latter is at its peak and the neuronal output has yet to be imposed. This would be the ideal time window to observe the effect of a TCA–secreted molecule, so it would be interesting to further discuss this possibility.

9. TCAs innervation's influence on cortical neurogenesis has been extensively correlated with the upper layer formation. The time of TCA arrival in the cortex closely matches the peak of upper layer neurogenesis, further supporting this view. It is quite surprising that the authors do not observe any alteration of any nature in the upper layers' analysis (Brn2 immunohistochemistry). Is this expected, or in line with other studies?

[Editors’ note: further revisions were suggested prior to acceptance, as described below.]

Thank you for resubmitting your work entitled "Thalamocortical axons control the cytoarchitecture of neocortical layers by area–specific supply of VGF" for further consideration by *eLife*. Your revised article has been evaluated by John Huguenard (Senior Editor) and a Reviewing Editor.

The manuscript has been improved, and Reviewers 1 and 2 applaud the very substantial work that you performed to improve the manuscript, with the clarifications and additional experiments you provide. Reviewer 4, who is new, calls it "a nice and important study". However, there are some remaining issues that need to be addressed, as outlined below:

Reviewer 1: In the last two comments, Reviewer 1 specifically asks for:

1. In Figure 2 figure supplement 3 E and G: this reviewer would like to see the same immunohistochemistry as performed in Figure 1 and 2 for 5HTT and *Rorb* expression. The labels are clearer ("labels" likely means "labeling".)

2. You state that the "the VB appeared intact in terms of shape, size, and cell density" (Figure 6B, 5 sections from 5 mice). Here quantification is requested to support this conclusion, given that there is a clear change in the shape of the VB nucleus.

Reviewer 4 asks a number of questions, the most interesting #2 and #5 regarding differences between primary sensory areas and motor cortex. These and the other comments can be addressed in the Discussion and also in your rebuttal letter, although the experiments that some of these points address would not be necessary to perform at this time.

*Reviewer #1:*

Overall, the authors have answered the main points of my previous review. However, some aspects of the imaging and data analysis could be further improved. The way of presenting the results is not well refined.

1) In Figure 1C, the VB nucleus shows the same size and shape in control and in the TCA–ablated model, and this is not consistent. I am still not fully convinced about the criteria for nuclei delimitation. It would have been convenient to use common markers for VB and dLGN as for example vGlut2.

2) In Figure 1 figure supplement 3B, the example image for the TCA–ablated is damaged in the nuclei in which the experiment is focused, therefore it should be replaced.

Lines 194–198: "DiI–labeled neurons that project to V1 were found in the dLGN and lateral posterior nucleus (LP) in both control and TCA–ablated mice (Figure 1—figure supplement 3A, B, 12 sections from 6 mice for each), and the ratios of retrogradely labeled cells with DiI in dLGN to those in LP were similar in the two groups." This is in my opinion an overstated statement based on the example that the authors show.

3) In Figure 2 figure supplement 3E and G, I would like to see the same immunohistochemistry as performed in Figure 1 and 2 for 5HTT and *Rorb* expression. The labels are clearer.

Lines 453–454: "Importantly, the thalamic structure was not affected: the VB appeared intact in terms of shape, size, and cell density (Figure 6B, 5 sections from 5 mice)." In my opinion, quantifications should be provided to support this conclusion. There is a clear change in the shape of the VB nucleus in the Vgf–/– example the authors provide.

*Reviewer #3:*

As I mentioned in the first round of reviewing, this is an important study linking the area-specific thalamocortical innervation to the differences in cytoarchitecture of the cortex. For me, this can be considered as one of the most important questions of developmental neurobiology - what are the mechanisms to shape the cortex for the area-specific cortical computational functions.

Overall, the authors answered most of the criticisms, or explained why they could not do it for all 3 reviewers.

Their response for all my specific points were to my satisfaction:

1. The authors considered my comment and performed a reasonable attempt to answer it. Response with the new data is fine.

2. The authors performed in situ hybridization of RORβ on flat-mount sections of control and TCA-ablated mice and presented it for the reviewer in Figure 4. This answers my question.

3. The authors observed that TCA terminals are segregated in a barrel-like fashion in VGF-KO S1 revealed by 5HTT staining (Figure 6B), but the cytoarchitectonic pattern formation in layer 4 neuron is affected. This answers my question.

4. Microglia distribution in layer 4 revealed by IbaI immunostaining was not changed in the TCA ablated specimens.

The authors also examined the expression of Brn2 and Ctip2 in addition to RORβ in EdU-labeled samples and showed that none of these cohorts among EdU-labeled cells was altered.

5. As requested the reduction of ROR expression is discussed in previous and the present study with TCA ablation.

6. The differences between Mezzera and López-Bendito, (2016) are discussed and it is speculated that the differences emerge because (i) only the output from thalamus were altered primarily in an acute manner and (ii) timing of cell death matters. I am content with the changes in discussion.

7. The response is reasonable and to my satisfaction.

8. The authors argue that VGF plays a role in maintenance of layer 4 cell number at postnatal stages, but not a neurogenic role at embryonic stages. This argument is reasonable in the light of the results.

9. The authors explain the less impact on upper cortical layer formation is due to the postnatal ablation of TCAs when upper layer neurogenesis is finished. This is reasonable.

Both of my minor technical comments were answered.

Overall, I find the study much improved and I am still very enthusiastic to see it published at eLife.

*Reviewer #4 (Recommendations for the authors):*

In the manuscript titled " Thalamocortical axons control the cytoarchitecture of neocortical layers by area–specific supply of VGF" by Sato et al., the authors investigated the reasons for the distinct laminar architecture among different cortical areas. They focused on layer 4, which is thicker in the primary sensory cortices but is thinner in the motor cortex. Previous studies had shown that TCAs play instructive roles to regulate area–specific properties in layer 4. In this study, the authors extended their previous study and used a variety of methods, including ablating VB neurons, performing in utero electroporation to overexpress genes in the cortex, and generating KO animals to demonstrate that TCAs secrete VGF to control the number of layer 4 neurons in the sensory cortices.

Overall, it is nicely organized with very interesting findings and the description of their findings was clear. The authors nicely demonstrated that TCA ablation leads to the reduction of layer 4 RORβ–positive cells. The most convincing results were from the Vgf KO mice: while the TCAs are still projecting to S1 and inducing the expression of Btbd3 (suggesting the presence of TCA activities), the layer 4 cell density is greatly reduced in the Vgf KO.

I have few suggestions:

1. In Figure 1—figure supplement 3, strong input from Po was found in the TCA ablated S1 ("in TCA–ablated mice, the retrogradely labeled VB was markedly reduced in size, but the Po was broadly and intensely labeled" (p9)). This agrees with the previous study (Pouchelon, 2014) showing that the genetic ablation of the VB at birth rewired Po projections to S1 layer 4 neurons and that respecified layer 4 neurons. This might be the reason for the reduction of *Rorb* expression and reduced cell density in layer 4 in the TCA ablated S1. I am wondering if the authors had considered this as one of the reasons for the phenotypes observed. Using additional S1–specific markers (such as MDGA1, as described in Takeuchi, 2007) might be able to clarify this point.

2. As in the introduction, the authors mentioned that the major differences in layer 4 neuron density are between sensory cortices and motor cortex. Did the authors electroporate VGF into layer 4 of the motor cortex or any other cortical areas (such as S2, next to S1) in control animals, or even in L5 or L2/3 in S1? I am wondering whether layer 4 neurons in sensory areas have area–specific responses to VGF (e.g. expression of the receptors).

3. The authors suggested the cell death could be a possible reason for the decrease of layer 4 neuronal number in TCA ablated S1, but it was difficult to show convincing data as it is not easy to detect dying cells. My suggestion is to compare the number of EdU+ cells (EdU injected at E14.3 to label layer 4 neurons), from P0 to P7. This way, one could detect whether layer 4 neuronal number is decreased after TCA ablation.

4. I am puzzled by the result presented in Figure 7—figure supplement–1H, where the authors showed among EdU+ cells in the Vgf mutants, while layer 4 *Rorb*+ cells are decreased, but others are not changed. They showed ~50% *Rorb*+, ~35% Brn2+ and ~2% Ctip2+ in wild type and ~30% *Rorb*+, ~30% Brn2+ and ~2% Ctip2+ in mutants. What other cell types are the rest of EdU+ cells in the mutants?

5. In Figure 7A, it seems like differences in layer 4 neuronal density could still be detected between primary sensory areas and motor cortex in the Vgf KO. It could be informative to compare the relative layer 4 neuronal density in S1 and motor cortex in WT and KO.

6. Do Vgf KO animals show a developmental delay? It would be informative to compare the distribution variance of layer 4 cells at a later time point between WT and KO.

---

## [Author Response]

[Editors’ note: The authors appealed the original decision. What follows is the authors’ response to the first round of review.]

Reviewer #1:Sato et al. investigated the role of the thalamus–derived factor, VGF, as extrinsic cue that controls layer 4 development in the cortex. They show that this process is necessary for further maturation of the primary somatosensory cortex (S1) and the barrel field formation. To explore the role of thalamocortical axon (TCA) projections in cortical layer formation, the authors developed a mouse model with TCA ablation from the ventrobasal (VB) thalamic nucleus by the administration of diphtheria toxin (DT) from P0. They induced a decrease in the VB nucleus size and the TCAs terminals were also diminished in layer 4 of S1. Sato et al. demonstrated that the number of layer 4 cells in S1 is reduced in TCA–ablated model and to verify that these TCA ablation from the VB was indeed responsible for this laminar phenotype, they used a different approach to specifically ablate only the VB neurons. By performing a technically impressive in utero electroporation at e11.5 in the thalamus with a diphtheria toxin receptor (DTR) expression plasmid and then administered DT at P0, they mimicked the laminar phenotype in the cortex and the results were similar to the TCA–ablated mice. Moreover, they showed that, apparently, the rest of the cortical layers remain intact. Interestingly, the authors demonstrated that VGF and NRN1 as TCA–derived factors play an important role to maintain the layer 4 neuronal number during cortical development by restoring these cell number after the overexpression of these factors in TCA–ablated model in vivo. To further address this question, they induced a genetic inactivation of Vgf by using CRISPR/Cas9–mediated gene editing, and they proved that VGF is necessary for the maintenance of layer 4 neurons number.The manuscript shows potentially interesting findings but there are some open questions and experiments that should be done to better support the conclusions of the paper. Moreover, some aspects of image acquisition and data analysis need to be clarified.1) The figures need to be reorganized to achieve a better flow of the manuscript. For example, Figure 1 does not contribute to the story, and it would make more sense to transfer it to supplementary figures. Moreover, the Figure 5—figure supplement 1 could be a principal figure to show the absence of VGF in thalamus and cortex of TCA–ablated mice because this leads to wonder about the role of this gene and justify the following experiments of overexpression and knock–down.

We reorganized the figures according to this suggestion. Previous Figure 1 and Figure 5—figure supplement 1 are now Figure 1—figure supplement 1 and Figure 4, respectively. The other figure numbers were adapted accordingly.

2) In Figure 2—figure supplement 1 the authors show lipophilic dyes tracings assuming that they only labeled TCAs and thalamic neurons retrogradely. However, this is a conceptual mistake because these tracers label indistinctly, retrogradely and anterogradely. They inject the dyes at P7 when corticothalamic axons have already innervated the thalamus and thus presumably, the labelling shown in PO is, at least in part, from S1 layer 5. It would be better to use a L4 specific virus to trace only retrogradely projections. Moreover, they state that the number of labeled cells were similar in TCA–ablation and control mice, however, there is a need to increase the n to support this statement as there is a strong variability in this technique due to the change in labelling related the change in the size of the dyes.

Thank you for this criticism. Indeed, our method do not distinguish signals labeled retrogradely or anterogradely, but we still can focus on the distribution of the labeled cell bodies by reducing an intensity threshold, which are mostly retrogradely labeled (anterograde labeling yields less intensive and diffuse signals). Magnified views of VB and PO were added to show retrogradely labeled cell bodies (Figure 1—figure supplement 3B, lower panels). We also described it in Results (P9, L4-6). The number of specimens was increased from 2 to 6 for both control and TCA-ablated mice, and we observed essentially same phenotype.

3) There are some affirmations in the manuscript that would need a quantification for support. For example, when the authors state that dLGN was not reduced in TCA–ablated mice would need a quantification. The same in Figure 2A where it is stated that 5HTT–Cre is expressed in dLGN, and in Figure 2D, where the authors show a reduction in the size of the nucleus. Therefore, dLGN size quantifications should be done and it would be interesting to show a slice of 5HTT–Cre;R26–YEFP in V1 layer 4 showing 5HTT–Cre expression at P0–P7.

We quantified the size of dLGN and VB (Figure 1E). In addition, we examined TCA projection from dLGN to V1 by 5HTT immunohistochemistry in P7 TCAablated mice. Although the staining intensity in V1 was weaker compared with S1, the amount of TCAs was comparable to that of control mice (Author response image 1) . This supports the notion that a substantial amount of dLGN neurons remained in 5HTT-Cre; R26-DTR mice upon DT administration.

**Author response image 1. sa2fig1:** Thalamocortical projection is preserved in V1 of TCAablated mice. Cross sections of V1 region of P7 control (upper panels) and TCA-ablated (lower panels) mice stained with anti-5HTT antibody. V1, primary visual area; L4, layer 4. Scale bar, 200 µm.

4) The results for cell death are not convincing, the authors should improve the detection method, for example, they could use caspase 3 in postnatal brains. In Figure 3—figure supplement 1 they only show quantifications in layer 2/3 and layer 4 where there is an increased tendency for cell death, but again the n is too small for some conditions. Additionally, in Figure 3—figure supplement 1C a quantification should be provided for cell death in layer 6. It is very likely that this cell death is due to the expression of Cre recombinase in layer 6 in the TCA–ablated mice. Thus, the results for cell death are not clear enough and some experiments should be repeated and /or conclusions clarify.

We did try several cell-death detection methods including caspase 3 (cleaved caspase 3, TUNEL, IbaI, ssDNA and mRNA of *Bax*, *Bad* and *Bak*) as described (P13, L15-22), and found that ssDNA immunostaining was the most sensitive way to detect dying cells in our situations. We now provided ssDNA data with increased N of upper layer (layer 2-4), deep layer (layer 5-6), and VB from P1 to P7 (Figure 2—figure supplement 2D). While induction of cell death was detected in deep layer and VB, we failed to observe increase of the signals in upper layer.

5) The authors sometimes underestimate the interpretation of the results. For example, in Figure 4 Tbr1 positive cells (layer 6) are reduced in TCA–ablated mice, and thus indicate that layer 6 is also affected in this mouse. Therefore, this model might not be the best one to investigate how extrinsic cues mechanisms regulate layer 4 development.

We think that the reduction of Tbr1-positive layer 6 cells is largely due to Cre expression by this population. We provided evidence that reduction of layer 4 neuron is solely due to TCA ablation by electroporating a DTR-expressing plasmid to E11.5 thalamus (Figure 2—figure supplement 3). We also ablated layer 6 neurons by electroporation to E11.5 cortex to see its effect on layer 4. Whereas Tbr1-positive layer 6 neurons were greatly reduced, the number of RORβ-positive layer 4 neurons was not markedly changed (Author response image 2) . We described this in Results (P14, L19-23).

**Author response image 2. sa2fig2:** Postnatal L6 ablation does not affect L4 cell number at P7. (A-C) Coronal sections of P7 cortices electroporated with DTR and DsRed at E11.5, when layer 6 neurons are produced. Without DT administration, DsRedpositive cells are mainly located in Tbr1-positive layer 6 at P7 (A). Upon DT administration at P0, the number of Tbr1-positive layer 6 cells was markedly reduced on electroporated (e.p.) side compared with control side at P7 (B). The number of RORC-immunoreactive layer 4 cells was not different between the hemispheres (C). (D) Quantification of the results. The number of RORC- and Tbr1-immunoreactive cells is presented as a percentage of control (mean ± SEM): Tbr1, 41.40 ± 6.44%; RORC, 104.25 ± 5.25%, N = 3 for each. L4, layer 4; L6, layer 6. Scale bars, 100 µm.

6) It would be interesting to explore the neural activity of layer 4 in the somatosensory cortex of Vgf–KO and TCA–ablated mice. Behavioral experiments with environmental enrichment and c–fos immunostaining would demonstrate the laminar functionality of the cortex under these different conditions.

Thank you for suggesting an interesting point. We have examined expression of c-Fos by immunostaining at P7 and found that the signal was hardly detected in the neocortex including S1 both in control and TCA-ablated mice, suggesting that the neuronal activity in the cortex is not yet very strong at P7. Since these issues are beyond the scope of the present study, we just would like to show Btbd3 data as evidence for neuronal activity.

Reviewer #2:Sato H. and colleagues here investigate the role of extrinsic factors in the development of the murine neocortex. It was previously shown by these authors that thalamocortical neurons, through the expression of Vgf and Nrn1 among others, contribute specifically to layer 4 neurons development in vitro. In the current study, by postnatally ablating thalamocortical projections and studying a Vgf knockout, they further investigate the role of these projections in layer 4 establishment and bring some light to the in vivo role of VGF.Although this is an interesting study, the novelty is relatively limited as it incrementally builds on previous work from this laboratory as well as previous work from several laboratories directly addressing the effects of ablation of specific thalamic nuclei on cortical neuron identity. In order to increase interest and relevance, several key experiments should be performed.– It is interesting to see that RORC staining emerges in a rostrocaudal manner from P0 to P7. With the current data, the authors could perhaps assess whether this emergence also occurs in a mediolateral manner. This would help to better describe the emergence of the layer 4 across cortical areas. Moreover, to confirm the statement of layer 4 being thicker in S1, a quantification or a reference to quantification of layer 4 across cortical areas would be required.

Thank you for the suggestion. We added a series of coronal sections in Figure 1—figure supplement 1B, showing that RORC immunostaining emerges in a lateral-medial (ventral to dorsal) fashion. Regarding a quantitative assessment, we analyzed the intensity of RORC immunostaining signal in Figure 1—figure supplement 1B to show clear boundaries of S1, especially PMBSF (posteromedial barrel subfield), become evident after P0 by P7. We added it as Figure 1—figure supplement 1C and described it in Results (P7, L4-9).

– Some of the quantifications and experiments present only 2 cases. The authors should consider at least to increase the number of cases to 3. In addition, considering the number of cases in most of the experiments, the use of parametric statistical analyses is incorrect (not possible to assess dataset normality with less than 30 data points). The authors should switch to non–parametric statistical analyses.

Thank you for the criticism and suggestion. We have increased the number of cases more than 3 with two exceptions listed below:

– Figure 4C: NRN1 immunostaining (n=2), because anti-NRN1 antibody (cat# s25261, Santa Cruz) was discontinued.

– Figure 2—figure supplement 2: P1 TCA-ablated mice (n=2), due to shortage of the animal resource.

We re-evaluated cases with < 30 data points using Mann-Whitney U test as a non-parametric statistical analysis. However, because the test needs paired data points more than 3 and 5 or 4 and 4, some experiments did not meet the criteria and could not perform a statistical analysis (Figure 2—figure supplement 2D, P1 to P5; Figure 4C, NRN1).

– The use of a Cre–dependent DT strategy to ablate thalamic neurons is interesting to study the role of their projections postnatally. However, it seems important to better analyze the dynamics of their death after DT administration. The authors should try to assess, both thalamic and cortical, cell death from 24h post DT administration up to their analysis time point (P7). It is important to understand the dynamics of cell death in both regions to try to establish a clearer correlation. This could by assessed by performing Caspase3 immunolabelling at P1, P2, P3, P4, P5, P6, and P7 after P0 DT injection both in the thalamus and the cortex. In Figure 5C, it is unclear why there is no Dd–red signal in the control and DTA conditions.

Thank you for the insightful comment. We provided the time course of the ssDNA data both for the thalamus (VB) and the cortex (L2-4 and L5-6, separately) from P1 to P7 (Figure 2—figure supplement 2D), showing that cell death in VB and L5-6 increased within several days after DT administration. We still could not detect a sign of cell death up-regulation in L2-4. We described it in Results (P13, L13-22). In Figure 5C, the control and TCA-ablated rows represent the non-electroporated side of the animals. We have confirmed that electroporation of Ds-Red only had no effect on L4 (data not shown).

– The authors should consider further describing the impact of this strategy on deep–layer neurons (YFP+ in Figure2A). As they are Cre+ during their developmental time course, DT is most likely going to target them (which could explain the reduced thickness of deep layers when compared to controls). This could be added as supplemental information, but it is important to evaluate the ground state of the neocortex to evaluate the real impact of TCAs.

Thank you for the suggestion. We analyzed cell death in deep layers from P1 to P7 after DT administration at P0. It peaked at P5-6 (Figure 2—figure supplement 2C, D), consistent with Cre activity in layer 6 (Figure 1A). In order to examine effects of cell death in layer 6 on layer 4, we electroporated DTR to the cortex at E11.5 when layer 6 neuron are produced. We found that the number of RORβ+ cells in layer 4 was not reduced significantly despite massive cell death in layer 6 (Author response image 2, Reviewer 1-point 5). Therefore, we conclude that the reduction of layer 4 is most likely due to elimination of TCA rather than cell death in layer 6. We mentioned this in Results (P14, L19-23).

– For the panels containing thalamic nuclei, it would help to have white dotted lines (as in Figure 2D) around them in all the panels. As an example, it is difficult to determine the MG position in DiA images (Figure 2—figure supplement 1).

We encircled the VB and MG in all panels containing thalamic nuclei.

– Throughout the paper, the fluctuation of layer 4 number of neurons throughout conditions is assessed by the expression of RORC (example: Figure 3C). In addition, to investigate possible layer 4 fate changes, the authors quantify L2/3 or L5/6 number of cells (with known markers). In order to improve the analysis of layer 4 neurons, instead of using postnatal markers, it would perhaps be interesting to electroporate at E14 to have permanent labelling of this cell population. On the one hand, it would help at first to assess if these specific cells are dying, and on the other, it would better allow the analysis of possible fate changes. This methodology would be complementary to the actual Cre–DTR system, making it optimal for this type of analysis. Another possibility would be to inject birth dating markers embryonically, such as CFSE.

Thank you for the suggestion. We have tried to label layer 4 neuron both by electroporation and CFSE before the initial submission. However, we concluded that these methods with our hands are not suitable for reliable quantitative statistical analyses, because variability of labeling among the specimens was considerably high. We then examined the possibility of fate change by labeling prospective layer 4 cells by EdU and examined the expression of RORC. The data strongly support that the cells themselves disappeared (Figure 2E, F; Figure 2—figure supplement 2A, B). We further examined Brn2 and Ctip2 expression, however we did not detect significant increase in the percentage of Brn2 or Ctip2-positive cells among EdU-positive cells (Figure 2—figure supplement 2A, B). We described it in Results (P12, L8-P13, L11).

– Regarding Nrn1 and Vgf rescue experiments, it is interesting to see that the absence of TCAs promotes their absence in layer 4. It would be interesting to add whether the over–expression of Nrn1 and Vgf, alone, can maintain "normal" levels of RorC+ cells in layer 4. Moreover, besides analyzing the presence of DsRed (Figure 5C) in double electroporations, it would be interesting to see the presence of the proteins themselves (either by ISH or immunolabelling) as a positive control for the experiment.

We provided expression of VGF and NRN1 in the electroporated specimens (Figure 5—figure supplement 1). We also conducted a solo rescue experiment with VGF or NRN1, and found that VGF but not NRN1 was sufficient to restore the phenotype (Figure 5E). Neither of them affected the number of layer 4 in the presence of TCA (data not shown). We described this in Results (P19, L1-7).

– Additionally, it would be interesting to visualize the expression of Vgf and Nrn1 receptors expression by layer 4 cells and their expression dynamics. Is it constantly expressed or there is a critical period for their expression? Can the authors rescue L4 neurons by overexpressing Vgf in L2/3 neurons? This would demonstrate a non–cell–autonomous effect onto L4 neurons.

It would indeed be interesting, but receptors for VGF and NRN1 are currently unknown. In some specimens for the rescue experiments in which exogenous VGF was expressed in the bottom part of layer 2/3, the layer 4 phenotype was restored, suggesting that VGF acts on layer 4 neurons in a non-cell autonomous fashion.

– In Vgf–/– mutants, there is an increase of Brn2+ and Ctip2+ cells, which could correspond to a fate change of layer 4 cells in absence of Vgf. It would be interesting to electroporate Gfp at E14 to label layer 4 cells and analyze the expression of these markers at P7.

Increase of Brn2+ and Ctip2+ cells in VGF-KO is not statistically significant (Figure 7G). We examined the possibility of fate change in *Vgf*-KO by EdU labeling at E14.3 which gives global and unbiased labeling of cells born immediately after as we did for the TCA-ablated specimens (Figure 2—figure supplement 2A, B). The result indicated decrease of RORC-positive cell population, but no substantial increase in Brn2- or Ctip2-positive cell populations among EdU-labeled cells (Figure 7—figure supplement 1G, H). Although we cannot rule out the possibility that the fate conversion occurred, our result at least argue against that fate change of layer 4-destined neurons is the major cause of the reduction of RORβ+ cells in layer 4. We explained it in Results (P24, L1-21).

– For Nrn1–/– analysis of layer 4 cell number, there should be a quantification. It is hard to tell from the images alone if there is a difference or not.

We increased the number of cases and added the quantification data to Figures (Figure 7—figure supplement 2B).

Reviewer #3:[…]Overall, this is an excellent study with important findings.1. While the authors clearly observed that layer 4 exhibited marked reduction in the number of neurons specifically within the primary somatosensory cortex and they reported no change in the number of layers 2–3, 5 or 6 neurons, I would be still interested in the neuron numbers in the entire depth of the cortex in an arbitrary unit column. "The number of RORβ–expressing cortical cells was decreased to 67% as compared with control cortex. Moreover, the absolute number of layer 4 neurons also decreased in the TCA–ablated S1." This argued against the possibilities of altered RORβ expression or fate change of the layer 4–destined cells to those of another layer. Is the reduction being just because the effect to layer 4?

We counted pan-neuronal marker NeuN-positive cells in an arbitrary unit column passing through layers 2-5 in S1. Layer 6 was excluded because Cre-positive cells are present and they would die regardless of TCA upon DT administration. NeuN+ cell number was slightly decreased in TCA-ablated mice, although it was not statistically significant (Figure 2—figure supplement 1A, C). It is perhaps due to a sort of dilution such that the decrease in layer 4 cell number was diluted by an overwhelming number of total NeuN-positive cells which include GABAergic neurons as well. Instead, we analyzed the number of Cux1-positive layers 2-4 neurons in S1 and found slight but significant decrease in TCA-ablated mice (Figure 2—figure supplement 1B, C). We described it in Results (P12, L17-20).

2. While the study is commenting on the changes within S1, it does not comment on the possibility that the size of S1 changed. It would be good to show the areal extension of the S1 barrel field on flatmounts and compare the representation of the S1 and V1 after all these manipulations. However, it is not essential for the basic conclusions of the study.

Thank you for pointing out the interesting point. We have tried assessment of the cortical areas with RORβ, Cad6, Cad8, Igfbp4, and Bhlhb5 on flat-mount sections at the plane of layer 4 as well as whole-mount cerebral hemispheres of P7 mice. However, the whole-mounts did not provide informative results. Instead, we conducted in situ hybridization of RORβ on flat-mount sections of control and TCA-ablated mice. Although the barrel pattern disappeared and areal borders became indistinct in TCA-ablated mice, as reported for another TCA-deficient mice (Olig3-Cre-specific Gbx2-cKO; Vue et al., 2013), the size and position of S1 barrel field including PMBSF (posteromedial barrel subfield) and ALBSF (anterolateral barrel subfield) and V1 were apparently similar between them (Author response image 3) . Since the analysis is not thorough enough, we would like to leave this issue out of the present study.

**Author response image 3. sa2fig3:** Area pattern on tangential sections of TCA-ablated mice. In situ hybridization for *Rorβ* on flat-mount sections of P7 control and TCAablated mice. PMBSF, posteromedial barrel subfield; ALBSF, anterolateral barrel subfield; V1, primary visual area; A1, primary auditory area; m, medial; a, anterior. Scale bar, 1 mm.

3. Formation of the barrels requires two stages. The periphery related sorting of the thalamocortical afferents (revealed with stains for thalamocortical afferents 5HTT or DiI or CO staining) and then, the activity dependent sorting of thalamocortical afferents will impose the cytoarchitectonic changes that will form the cytoarchitectonic barrels (revealed with DAPI, Nissl or NeuN staining) (see Lopez–Bendito and Molnar, 2003). It would be good to clarify which mechanism is affected.

Thank you for the suggestion. We did observe that TCA terminals are segregated in a barrel-like fashion in VGF-KO S1 revealed by 5HTT staining (please see Figure 6B). Thus, it seems that cytoarchitectonic pattern formation in layer 4 neuron is affected. We added this point to the text (P25, L4-6; P33, L11-14).

4. The authors suggested cell death in layer 4 as a possible mechanisms for getting these differences. However, all the conventional cell death detection methods (ssDNA, cleaved caspase 3, Iba1, mRNA of Bax, Bad, and Bak, and DAPI), they could not obtain convincing evidence for significant cell death induction in layer 4 upon TCA ablation. Was there any change in microglia distribution in layer 4 after the ablation of the VB thalamic neurons?Even though the authors justify a possible technical difficulty in detecting dead cells in a dynamic developing system (which I completely agree with and understand), they don't provide a solid explanation for such a great decrease in cell number in layer 4. As the authors describe layer 4 neurons are already generated at the time of ablation, therefore they either die or change their fate. The cell fate change should be better investigated and discussed. Can the loss of 67% of neurons be completely due to cell death?

Thank you for the insightful comment. We observed that microglia distribution in layer 4 revealed by IbaI immunostaining was not changed in the TCA ablated specimens. To investigate whether fate change of layer 4 neurons is involved in TCA-ablation phenotype, we examined the expression of Brn2 and Ctip2 in addition to RORβ in EdU-labeled samples (Figure 2—figure supplement 2A, B). None of these cohorts among EdU-labeled cells was substantially altered. Thus, at this moment we at least can conclude that cell fate change is not the major cause of the reduction of layer 4 cells in TCA-ablated mice, although we cannot exclude the possibility of fate conversion of layer 4-destined cells. We mentioned this in the text (P12, L21-P13, L11).

5. There are numerous studies that demonstrated that six layered cortex can develop relatively normally without thalamic connections (Miyashita–Lin et al., 1999 DOI: 10.1126/science.285.5429.906; Zhou et al., 2010, DOI: https://doi.org/10.1523/JNEUROSCI.6005–09.2010). Did any of these studies suggested reduced layer 4 cell numbers? Do the authors used more sensitive methods for their detection? These issues should be discussed in more detail.

Those authors analyzed ROR expression by in situ hybridization. While the number of neurons in each layer was not counted in the previous studies, reduction of ROR expression is consistent with the present study. We mentioned this in Discussion (P29, L19-22).

6. The study described that after ablation at P0, VB was significantly reduced by P5. Parallel with this RORα–expressing thalamic neurons were also decreased. The traced from S1 and did not see any signs of TCA rewiring from the dLGN or MG to S1. Why do the authors think that Mezzera and López–Bendito, (2016) had different findings? Is is because of the methods, timing, tracing?

Thank you for bringing up the important point. We assume that rewiring may occur when inputs to the thalamus have been changed. We speculate that we could not detect rewiring because only the output from thalamus were altered primarily in an acute manner. Also, we think that the timing of cell death matters: Pouchelon et al., (2014) reported that using a DTA-expressing strategy, which should lead to cell death at earlier time point than ours, rewiring from PO to S1 occurred. We described this point in Discussion (P29, L22-P29, L6).

7. The results explain why layer 4 is thick in the somatosensory cortex. However, it is not known whether the same mechanism operates in the motor cortex where there is a very thin layer 4. The authors reduce the possibility of a cell death–based mechanism by examining the Bax and Casp3 mutant mice. In this lone the laminar configuration across the areas appeared to be normal. Do the authors suggest that the thinner layer 4 in M1 is the result of a different neurogenetic program generating these proportions?

In the present study, we have not addressed experimentally why layer 4 in M1 is thin. Yet, the results obtained from *Bax*- and *Casp3*-KO mice suggest that it may not depend on cell death, exactly as the reviewer pointed out. At present, we would like just to mention this point in Discussion (P30, L16-21; P31, L1015), but to leave this issue for future studies.

8. Do the authors suspect possible interaction with neurogenesis if the VB is ablated from early embryonic stages? I am not requesting this experiment, just some speculations in the discussion.In fact, TCA ablation at earlier periods of development is expected to have a greater effect on neurogenesis, as the latter is at its peak and the neuronal output has yet to be imposed. This would be the ideal time window to observe the effect of a TCA–secreted molecule, so it would be interesting to further discuss this possibility.

Thank you for bringing up the important point. Although it is reported that TCAs affect cortical neurogenesis (e.g., Gerstmann et al., 2015; Kraushar at el., 2015), such effect may not be restricted to sensory TCAs. We so far have not found obvious neurogenic abnormality in Vgf-KO cortex at P7 (Figure 7E-G) and also at P0 (Figure 7—figure supplement 1C-F, newly added data). These results suggest that VGF plays a role in maintenance of layer 4 cell number at postnatal stages, but not a neurogenic role at embryonic stages. Although we may be able to eliminate TCAs at the earlier time point, it would be out of scope of the current study (i.e., layer 4 in sensory areas and TCA innervation). We discussed this point in Discussion (P30, L6-15).

9. TCAs innervation's influence on cortical neurogenesis has been extensively correlated with the upper layer formation. The time of TCA arrival in the cortex closely matches the peak of upper layer neurogenesis, further supporting this view. It is quite surprising that the authors do not observe any alteration of any nature in the upper layers' analysis (Brn2 immunohistochemistry). Is this expected, or in line with other studies?

We interpret that the less impact on upper layer formation is due to the postnatal ablation of TCAs when upper layer neurogenesis is almost finished. In that sense, VGF does not appear to play a role in upper layer neurogenesis. We mentioned this point in Discussion (P30, L6-15).

[Editors’ note: what follows is the authors’ response to the second round of review.]

Reviewer #1:Overall, the authors have answered the main points of my previous review. However, some aspects of the imaging and data analysis could be further improved. The way of presenting the results is not well refined.1) In Figure 1C, the VB nucleus shows the same size and shape in control and in the TCA–ablated model, and this is not consistent. I am still not fully convinced about the criteria for nuclei delimitation. It would have been convenient to use common markers for VB and dLGN as for example vGlut2.

In Figure 1C, the size of the VB has not yet become markedly reduced, as cell death just started and would proceed rapidly thereafter (please see Figure 2—figure supplement 2D). In fact, the size of the VB is quite fluctuated among the P5 specimens. The VB size was much reduced consistently at P6 or later, as shown in Figure 1D. We replaced Figure 1C with the one with more shrunk VB.

2) In Figure 1 figure supplement 3B, the example image for the TCA–ablated is damaged in the nuclei in which the experiment is focused, therefore it should be replaced.

We replaced the panels with those without damage.

Lines 194–198: "DiI–labeled neurons that project to V1 were found in the dLGN and lateral posterior nucleus (LP) in both control and TCA–ablated mice (Figure 1—figure supplement 3A, B, 12 sections from 6 mice for each), and the ratios of retrogradely labeled cells with DiI in dLGN to those in LP were similar in the two groups." This is in my opinion an overstated statement based on the example that the authors show.

We admit that this particular image does not necessarily support this notion. We removed this statement which it is not essential for the main conclusion.

3) In Figure 2 figure supplement 3E and G, I would like to see the same immunohistochemistry as performed in Figure 1 and 2 for 5HTT and Rorb expression. The labels are clearer.

We performed another immunohistochemistry using the same staining method (fluorescence) so that Figure 2 figure supplement 3E, F appear in a similar way to Figure 1 and 2 (the former F and G are integrated as F).

Lines 453–454: "Importantly, the thalamic structure was not affected: the VB appeared intact in terms of shape, size, and cell density (Figure 6B, 5 sections from 5 mice)." In my opinion, quantifications should be provided to support this conclusion. There is a clear change in the shape of the VB nucleus in the Vgf–/– example the authors provide.

This particular image shows the VB in a different shape perhaps due to a plane of section, but its overall shape was similar between the mutant and wild-type animals. Nonetheless we deleted “shape” from the text and replaced the image of a section in which the VB appears more similar to the control. We measured the area size of the VB in sections where it is represented at the maximum. The result is provided as Figure 6C.

Reviewer #4:In the manuscript titled " Thalamocortical axons control the cytoarchitecture of neocortical layers by area–specific supply of VGF" by Sato et al., the authors investigated the reasons for the distinct laminar architecture among different cortical areas. They focused on layer 4, which is thicker in the primary sensory cortices but is thinner in the motor cortex. Previous studies had shown that TCAs play instructive roles to regulate area–specific properties in layer 4. In this study, the authors extended their previous study and used a variety of methods, including ablating VB neurons, performing in utero electroporation to overexpress genes in the cortex, and generating KO animals to demonstrate that TCAs secrete VGF to control the number of layer 4 neurons in the sensory cortices.Overall, it is nicely organized with very interesting findings and the description of their findings was clear. The authors nicely demonstrated that TCA ablation leads to the reduction of layer 4 RORβ–positive cells. The most convincing results were from the Vgf KO mice: while the TCAs are still projecting to S1 and inducing the expression of Btbd3 (suggesting the presence of TCA activities), the layer 4 cell density is greatly reduced in the Vgf KO.I have few suggestions:1. In Figure 1—figure supplement 3, strong input from Po was found in the TCA ablated S1 ("in TCA–ablated mice, the retrogradely labeled VB was markedly reduced in size, but the Po was broadly and intensely labeled" (p9)). This agrees with the previous study (Pouchelon, 2014) showing that the genetic ablation of the VB at birth rewired Po projections to S1 layer 4 neurons and that respecified layer 4 neurons. This might be the reason for the reduction of Rorb expression and reduced cell density in layer 4 in the TCA ablated S1. I am wondering if the authors had considered this as one of the reasons for the phenotypes observed. Using additional S1–specific markers (such as MDGA1, as described in Takeuchi, 2007) might be able to clarify this point.

Thank you for an insightful suggestion. While neurons in the VB were eliminated slightly later than the case reported by Pouchelon et al., we observed that the PO was labeled more in some TCA-ablated specimens, suggesting that rewiring from PO to S1 may have occurred to some extent. Although we did not examine whether S1 is respecified to S2 in our case, we would argue that this could not be the primary cause of the phenotype, because overexpression of VGF rescued the phonotype. Moreover, Pouchelon et al. reported that the neuron number in S1 including layer 4 was unchanged, suggesting conversion to S2 does not accompany reduction of layer 4. We mentioned rewiring from PO in Discussion (page 28, line 22-page 29, line 3).

2. As in the introduction, the authors mentioned that the major differences in layer 4 neuron density are between sensory cortices and motor cortex. Did the authors electroporate VGF into layer 4 of the motor cortex or any other cortical areas (such as S2, next to S1) in control animals, or even in L5 or L2/3 in S1? I am wondering whether layer 4 neurons in sensory areas have area–specific responses to VGF (e.g. expression of the receptors).

As we mentioned in Discussion, the present notion does not necessarily explain why layer 4 is thin in motor cortex. Although VGF receptor is currently unidentified, it may not be expressed in the motor cortex or other layers. Given that the exogenous VGF did not show an additive effect in the presence of TCAs in S1 layer 4, we assume that effects on other layers in S1 would be less likely. On the other hand, it appears that VGF exerts its function widely in the sensory areas which exhibit various degree of layer 4 enrichment though, as layer 4 in V1 was also reduced in VGF-KO. Our preliminary trial by which VGF was electroporated in the motor cortex indicated that ROR-β-positive layer 4 was unchanged. Moreover, our observation of Casp3- and Bax-KO mice suggests that cell death may not be involved in the layer 4 formation in the motor cortex. We however would like to leave this issue for the future studies as we wrote in Discussion.

3. The authors suggested the cell death could be a possible reason for the decrease of layer 4 neuronal number in TCA ablated S1, but it was difficult to show convincing data as it is not easy to detect dying cells. My suggestion is to compare the number of EdU+ cells (EdU injected at E14.3 to label layer 4 neurons), from P0 to P7. This way, one could detect whether layer 4 neuronal number is decreased after TCA ablation.

We observed that EdU-labeled cells began to decrease at P4 and were reduced as much as P7 at P6, although the number of cases obtained is too small for statistic evaluation.

4. I am puzzled by the result presented in Figure 7—figure supplement–1H, where the authors showed among EdU+ cells in the Vgf mutants, while layer 4 Rorb+ cells are decreased, but others are not changed. They showed ~50% Rorb+, ~35% Brn2+ and ~2% Ctip2+ in wild type and ~30% Rorb+, ~30% Brn2+ and ~2% Ctip2+ in mutants. What other cell types are the rest of EdU+ cells in the mutants?

Thank you for pointing out an interesting issue. Currently we do not know the identity of those populations, although it is likely that they failed to express those layer markers examined, implicating a role of VGF in fate specification of the cortical neurons.

5. In Figure 7A, it seems like differences in layer 4 neuronal density could still be detected between primary sensory areas and motor cortex in the Vgf KO. It could be informative to compare the relative layer 4 neuronal density in S1 and motor cortex in WT and KO.

Thank you for an interesting suggestion. We added quantification of ROR-β expressing cells in M1 in Figure 7C. M1 was not statistically different between the mutant and wild-type animals. We would like to leave the further issue of motor cortex for the future study as mentioned above.

6. Do Vgf KO animals show a developmental delay? It would be informative to compare the distribution variance of layer 4 cells at a later time point between WT and KO.

Thank you for raising an interesting possibility. We searched for specimens of the adult homozygote to examine this possibility, but we did not have any unfortunately. Since it will take some time to accomplish this, we would be grateful if we may leave this issue at this time.